# Molecular architecture of the augmin complex

Clinton A. Gabel [1,2], Zhuang Li[1,2], Andrew G. DeMarco[2,3], Ziguo Zhang [4], Jing Yang[4], Mark C. Hall [2,3], David Barford [4] & Leifu Chang [1,2] ✉

Accurate segregation of chromosomes during mitosis depends on the correct assembly of the mitotic spindle, a bipolar structure composed mainly of microtubules. The augmin complex, or homologous to augmin subunits (HAUS) complex, is an eight-subunit protein complex required for building robust mitotic spindles in metazoa. Augmin increases microtubule density within the spindle by recruiting the γ-tubulin ring complex (γ-TuRC) to pre-existing microtubules and nucleating branching microtubules. Here, we elucidate the molecular architecture of augmin by single particle cryo-electron microscopy (cryo-EM), computational methods, and crosslinking mass spectrometry (CLMS). Augmin's highly flexible structure contains a V-shaped head and a filamentous tail, with the head existing in either extended or contracted conformational states. Our work highlights how cryo-EM, complemented by computational advances and CLMS, can elucidate the structure of a challenging protein complex and provides insights into the function of augmin in mediating microtubule branching nucleation.

The spindle apparatus is conserved across eukaryotes allowing for the congression and segregation of duplicated chromosomes during mitosis and meiosis. This fundamental structure is vital in cell division, and subsequently reproduction, growth, and development. Composed of highly dynamic microtubules in a bipolar array, the spindle must assemble and disassemble in an orchestrated manner to correctly separate duplicated chromosomes into two new daughter cells. Inaccurate segregation of chromosomes often leads to aneuploidy, a cause of cell death and malignancy. Therefore, the development of the spindle through microtubule polymerization is tightly regulated by many different proteins. Previous studies have revealed three distinct pathways for microtubule nucleation for spindle development: centrosome-based, chromosome-based, and intraspindle (microtubule branching) pathways[1]. Microtubule branching depends upon the function of the augmin complex, or the homologous to augmin subunits (HAUS) complex (hereafter augmin)[2–6]. Current models suggest that augmin binds to pre-existing microtubules within the spindle and recruits the γ-tubulin ring complex (γ-TuRC) via the neural precursor

cell expressed developmentally down-regulated protein 1 (NEDD1)[7–10] (Fig. 1a). This allows for the nucleation of shallow angled, daughter microtubules within the spindle to maintain spindle polarity. These branching microtubules propagate a denser network necessary for the efficient capture of kinetochores on sister chromatids. Along with augmin and γ-TuRC, the targeting protein for Xklp2 (TPX2) is also necessary for branching and is considered part of the minimal machinery essential for microtubule branching[4,11–13].

Apart from its role in spindle assembly, augmin plays an important role in neuronal migration, development, and polarization by locally regulating microtubule nucleation events; furthermore, augmin ensures uniform microtubule polarity in axons, axonal nucleation, and the regulation of microtubule density in dendrites[14–17]. Moreover, knockout studies of augmin subunits show spindle defects and mitotic delay in neural stem cells, leading to apoptosis and failure of brain development[18]. In plants, augmin has been shown to be a regulator in cortical microtubule organization[19] and organizing the spindle and phragmoplast microtubule arrays[20].

---

[1]Department of Biological Sciences, Purdue University, West Lafayette, IN 47907, USA. [2]Purdue University Center for Cancer Research, Purdue University, West Lafayette, IN 47907, USA. [3]Department of Biochemistry, Purdue University, West Lafayette, IN 47907, USA. [4]MRC Laboratory of Molecular Biology, Cambridge CB2 0QH, UK. ✉e-mail: lchang18@purdue.edu

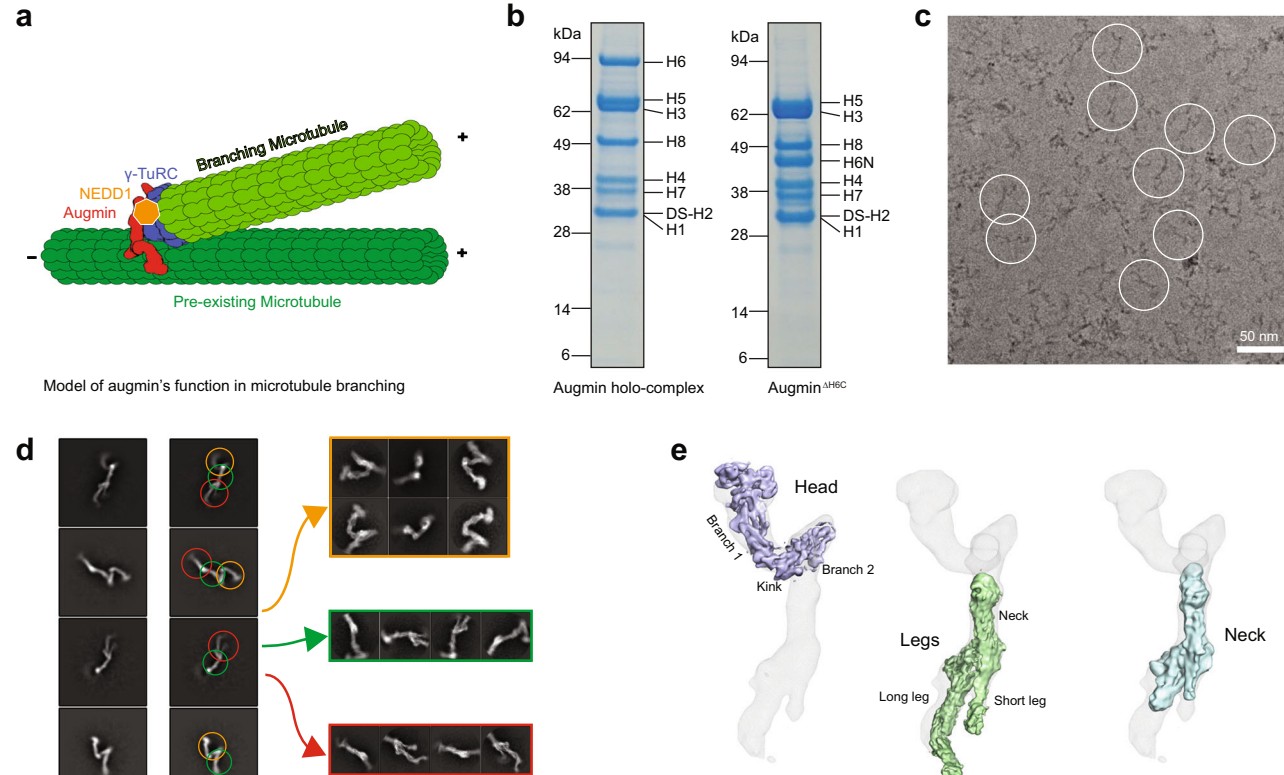

**Fig. 1 | Purification and cryo-EM reconstruction of augmin. a** A diagram of augmin–γ-TuRC–NEDD1 binding to pre-existing microtubules to create shallow-angled branching microtubules. **b** Representative SDS–PAGE gels of augmin holo-complex and augmin^ΔH6C. The results shown are representative of more than three experiments. The uncropped gel is provided in the Source Data file. **c** Representative cryo-EM micrograph of augmin^ΔH6C from a total of 20,021 micrographs. Scale bar 50 nm. Examples of augmin particles are circled. **d** 2D classification of augmin^ΔH6C where classes are segmented into V-shaped head (orange arrow), neck of the tail (green arrow), and legs of the tail (red arrow) before further 3D classification and refinement. **e** Cryo-EM maps of segmented regions including the V-shaped head (light blue), the legs (green), and the neck (cyan), superimposed to a low-resolution reconstruction of the whole complex (gray mesh).

Originally discovered in *Drosophila melanogaster* S2 cells, augmin was subsequently discovered in humans[21–23] and plants[20,24]. Augmin consists of eight subunits named HAUS1–8 (hereafter, H1–8), composed mostly of α-helices (Supplementary Fig. 1a). Functional analysis of different subunits, such as H1[25] and H8[26–28], indicated the importance of augmin in mitosis, evidenced by loss of HAUS proteins causing chromosome misalignment, multi-polar spindle asters, and cytokinesis failure[2,21,23]. In vitro reconstitution and negative stain EM data of human augmin revealed that augmin exists as an extended Y-shaped complex[29]. Additionally, reconstitution of *Xenopus laevis* augmin suggests that augmin adopts a similar shape in *X. laevis* compared to humans[30]. Recently, biochemical reconstitution of branching microtubule nucleation using purified components was achieved in both *X. laevis*[11] and *D. melanogaster*[13]. Despite these advances in isolating augmin and biochemical studies, a three-dimensional structure of augmin remains elusive, hindering mechanistic knowledge of how branching microtubule nucleation occurs.

To understand the molecular mechanisms of this process, we utilized single particle cryo-EM and advances in computational methods of protein structure prediction to build an atomic model of augmin.

## Results

### Cryo-EM of augmin

We prepared two complexes of human augmin octamers, the holo-complex and an octamer with the C-terminal region of H6 deleted (H6C, amino acids (a.a.) 433–955), leaving the N-terminal region (H6N, a.a. 1–432). We call this truncated octamer augmin^ΔH6C. Both complexes were prepared using the baculovirus-insect cell expression system[31] (Fig. 1b); however, truncation of H6C improved augmin complex expression and purification yield, consistent with a previous study[29]. Negative stain EM and cryo-EM images with 2D classification of both complexes (Fig. 1c, d and Supplementary Fig. 1b, c) showed a similar Y-shaped structure as previously reported[29,30], which we split into a V-shaped head and a tail. As H6C is mainly composed of unstructured loops, based on structure predictions (Supplementary Fig. 1a), and augmin^ΔH6C showed an indistinguishable structure when compared to the holo-complex, augmin^ΔH6C was utilized for cryo-EM.

Due to the flexible and elongated shape of augmin, we overcame several obstacles to reconstruct the cryo-EM structure of augmin. First, we used a thin, continuous carbon film to observe monodisperse Y-shaped particles, which otherwise were not seen on grids with no support or graphene oxide flakes in our study (Supplementary Fig. 1d–f). Due to the low contrast of augmin in cryo-EM images (particles were barely visible even at −5 μm defocus), a Volta Phase Plate was employed for data collection to increase image contrast. Second, we segmented the head and the tail regions for separate image analysis to reduce the degree of heterogeneity using an in-house script (Fig. 1d). This improved the resolution from ~15 to 5–7 Å (Fig. 1e, Supplementary Fig. 2, and Supplementary Table 1), allowing direct observation of α-helices (Fig. 2a–c). Interestingly, 3D classification of the head revealed two conformations: one extended (extended state) and the other compact (contracted state) (Supplementary Fig. 2d). The extended state of the head was more ordered and determined to a higher resolution, thus the extended state will be discussed below unless otherwise specified.

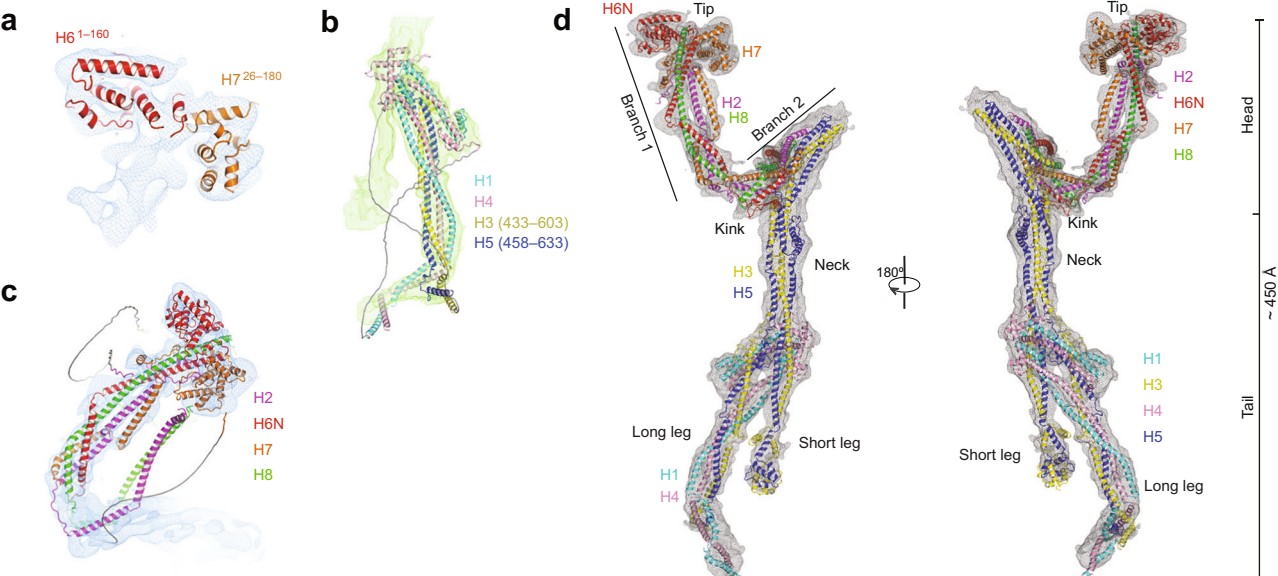

**Fig. 2 | Fitting of HAUS subunits and full atomic model of augmin. a** Rigid-body fitting of the individual AlphaFold2 predictions of H6 (a.a. 1–160) and H7 (a.a. 26–160) within the cryo-EM map of the tip of branch 1. **b** Rigid-body fitting of the ColabFold prediction of the long leg. **c** Rigid-body fitting of the ColabFold prediction of branch 1 of the V-shaped head. **d** Atomic model of augmin superimposed to a combined cryo-EM map (in gray mesh).

Overall, augmin adopts a ~45 nm long, highly flexible structure, containing a V-shaped head and a filamentous tail (Figs. 1e and 2d). The V-shaped head contains two branches at a ~50° angle. Branch 1, which is distal to the tail, has a length of ~160 Å and contains a globular domain at the tip measuring ~80 Å, *by* ~40 Å, *by* ~40 Å in dimension, whereas branch 2 is connected to the tail (Figs. 1e and 2d). The filamentous tail measures ~350 Å and is composed of a neck and two legs (one long and the other short).

## Subunit assignments

With recent advances in protein structure prediction, specifically AlphaFold2[32–34], we hypothesized that predicted subunit structures might fit into our cryo-EM maps. First, we attempted to fit single protein predictions from the AlphaFold Protein Structure Database[32,35]. The predicted structures of H1–8 individual subunits are in general long helices except for two small globular domains at the N-termini of H6 (a.a. 1–160) and H7 (a.a. 26–180) (Supplementary Fig. 3a). These two globular domains were fit with confidence into the globular density at the tip of branch 1 (Fig. 2a and Supplementary Fig. 3b–d). However, other regions and subunits could not be confidently fitted.

Second, following the release of ColabFold[33], which enables the prediction of protein complexes, we tried predicting the structures of all binary combinations of HAUS subunits (Supplementary Fig. 4a). This confirmed the interaction of the N-termini of H6 and H7 identified from fitting individual predicted models. In addition, this approach identified highly probable binary interactions of H2/6, 2/7, 2/8, and 6/8 (Supplementary Fig. 4a, b). Together, these predictions of binary interactions and their 3D structure indicate that the head region of augmin is likely composed of H2/6/7/8, consistent with previous studies showing these subunits form a tetramer[29,30]. Strikingly, two strong binary complexes were identified in the predictions: H1/4 and H3/5 (Supplementary Fig. 4a, b). H1/4 fitted almost perfectly to the long leg of the tail as a rigid body (Supplementary Fig. 5a), leaving some unassigned density in the long leg most likely contributed by H3/5 (discussed below). The N-termini of H3/5 match the short leg, whereas the middle region of the dimer fits to the neck (Supplementary Fig. 5b). In addition, interactions for H1/3 and H1/5 were identified (Supplementary Fig. 4a, b), suggesting the four subunits (H1/3/4/5) are located in the tail, consistent with a previous study showing these subunits form a tetramer[30].

Next, we attempted to predict structures with more than two subunits. Given a sequence length limit of ColabFold (<1400 a.a.), we made use of the structural information obtained from binary runs to group sequences. We assigned the extra density in the long leg to the C-termini of H3 (a.a. 433–603) and H5 (a.a. 458–633), which together with H1 and H4 form a stable unit (Fig. 2b and Supplementary Fig. 5c). The short leg is assigned to N-termini of H3 (a.a. 1–121) and H5 (a.a. 1–98), and the neck to the remaining parts of H3 and H5 (Supplementary Fig. 5d). By grouping sequences of H2/6/7/8, structures of branch 1, branch 2, and the kink between them were also predicted by ColabFold and fitted within the cryo-EM map of the V-shaped head (Fig. 2c and Supplementary Fig. 5e–g).

Taken together, using these predictions, we were able to build the intact augmin model in COOT[36] (except for a.a. 433–954 of H6 and a.a. 1–138 of H8), and assigned all subunits to their respective densities (Fig. 2d).

After building our model, AlphaFold Multimer was released[32,34]. AlphaFold Multimer predicted both branches of the V-shaped head and generated a structure where branch 2 is folded back onto branch 1 (Supplementary Fig. 5h), instead of a ~50° separation as observed in the cryo-EM density. Similarly, AlphaFold Multimer predicted the two legs and the neck, but the neck was folded back to the legs (Supplementary Fig. 5i). Taken together, cryo-EM paired with AlphaFold2 was found to be a powerful approach for building the structure of augmin.

## Crosslinking mass spectrometry

To independently assess the validity of our 3D structural model, we used crosslinking mass spectrometry (CLMS) to identify peptide pairs in close proximity within purified augmin[ΔH6C]. CLMS is becoming a useful tool for supplementing protein structural characterizations[37]. We treated purified augmin[ΔH6C] with disuccinimidyl sulfoxide (DSSO) at different molar ratios to primarily crosslink neighboring lysine amino groups and then digested the products with trypsin either directly in solution or after separation by SDS–PAGE (Supplementary Fig. 6a). Resulting peptides were subjected to liquid chromatography–mass spectrometry (LC–MS)

and the data analyzed using the MetaMorpheus CLMS search program[38]. Sequence coverage of augmin subunits ranged from 67.6% to 99.7% (Supplementary Dataset 1). We detected 33 unique crosslinks between augmin subunits (intersubunit crosslinks, Fig. 3 and Supplementary Fig. 6b, c, and Table 1) and 42 unique crosslinks within individual augmin subunits (intrasubunit crosslinks, Supplementary Fig. 7 and Supplementary Table 2) using search criteria that yielded ~1% crosslink FDR (see the "Methods" section). The reliability of MetaMorpheus in reporting true crosslinked peptide pairs is strongly supported by the fact that most identified intrasubunit crosslinks are neighboring lysine residues in the primary sequences, as would be expected, regardless of the folded structure. The intersubunit crosslink pairs provide the most valuable test of our structural model. Among the intersubunit crosslinked amino acid pairs, 25 of 33 have Cα–Cα distances below 23 Å. This is consistent with previously observed Cα–Cα distances of DSSO-induced crosslinks in proteins of known structure[39], and the predicted theoretical limit to Cα–Cα distance for the closely related DSS crosslinker is 26–30 Å[40]. An additional five intersubunit crosslinks contain one peptide that is not resolved from the cryo-EM map, presumably due to flexibility. However, in these cases the missing peptide would still be in near proximity to its linked partner peptide (<50 Å) (Fig. 4c). Only 3 of the 33 intersubunit crosslinks are inconsistent with the model based on predicted Cα–Cα distances (crosslinks 18 (54 Å), 12 (59 Å), and 19 (312 Å) in Fig. 3c and Table 1). The few inconsistencies could reflect complex flexibility and different conformational states of augmin, crosslinks between multiple augmin complexes, or false positive search matches. Multiple intersubunit crosslinks were detected in the V-shaped head, the neck, and the legs of augmin^ΔH6C. Collectively, the CLMS data support the overall accuracy of our atomic model of augmin, although some regions of the complex lacked CLMS data coverage.

### Structure of the V-shaped head

Overall, augmin adopts an elongated shape composed of inter-connecting, helical bundles (HBs) (Fig. 2d). The V-shaped head consists of a series of HBs starting at the tip of branch 1 and continuing through the kink and up into branch 2 (Fig. 4a–c). H2/6N/7/8 (consisting of 7, 13, 13, and 3 helices, respectively) run in parallel, with their N-termini located at the tip of branch 1 and C-termini located in branch 2 (Fig. 4d–g). The tip of branch 1 is composed of the N-termini of H6 (α1–8 and N-terminal region of α9), H7 (α1–8), H2 (α1–2), and H8 (N-terminal region of α1). After the tip, the rest of branch 1 and branch 2 consists of four consecutive HBs, formed by inter-twisting helices from all four subunits. Branch 1 can be split into HB1 (H2-α3, H6-α9, H7-α10, and H8-α1) and HB2 (H2-α4, H6-α10, H7-α11, and H8-α1) with α1 of H8 spanning both HBs (Fig. 4b). Branch 2 contains HB3 (H2-α7, H6-α13, H7-α13, and H8-α3), connected to branch 1 through a kink made of a six-helix bundle (α5–6 of H2, α11–12 of H6, α12 of H7, and α2 of H8) (Fig. 4b). At the end of branch 2 is another HB from H3/5 (HB4) to be discussed below (Fig. 5c). Of note, a.a. 1–138 of H8 are not observed in the cryo-EM map, consistent with circular dichroism (CD) spectrometry data which indicate that this region is a random coil in solution[29]. Additionally, the C-terminus of H8 (a.a. 340–410) was also a random coil as predicted by AlphaFold2 with no corresponding density within our cryo-EM map.

In the contracted state of the head, both branches 1 and 2 (~100 and ~90 Å in length, respectively) are shorter than in the extended state (~160 and ~110 Å, respectively), while the angle between the two branches remains similar in both states (Fig. 4h). Branch 1 in the contracted state was poorly resolved in the cryo-EM map possibly due to structural instability, flexibility, or both, preventing accurate modeling. In branch 2, HB4 from H3/5 exhibits a ~20 Å shift towards the kink from the extended to the contracted state, accompanied with

rotations in the kink and HB3 of the head (Fig. 4i, j and Supplementary Movie 1).

### Structure of the tail

Sitting beneath the V-shaped head is the neck, a domain that forms the upper part of the tail. Below the neck, the complex splits into short and long legs (Fig. 5a, b). The neck, composed of H3 and H5, each containing 15 helices apiece, is segmented into three HBs, HB4 (H3-α8–9 and H5-α9–10), HB5 (H3-α7,10 and H5-α8,10), and HB6 (H3-α6,10 and H5-α5,10) (Fig. 5c–e). HB4 forms the connection between the head and tail of the complex (Fig. 4b), whereas H5-α10 links all three HBs within the neck. Additionally, HB4 exhibits the ~20 Å shift toward the kink in the contracted state as mentioned above (Fig. 4i, j and Supplementary Movie 1). A distinctive feature of the neck is a knob, created by H5-α6–7, which demarks the center of the neck and sits on the periphery of HB5 (Fig. 5c–e). Next, H3-α6 and H5-α5 reach into the short leg formed by the N-termini of H3/5, while disordered loops (a.a. 428–445 of H3 and a.a. 452–477 of H5) following H3-α10 and H5-α10 designate the transition from the neck into the long leg formed by the C-termini of H3/5 and a stable H1/4 dimer.

The legs in the tail begin at the bifurcation point where the H1/4 dimer starts interacting with the H3/5 dimer (Fig. 5a). In the short leg, H3-α6 and H5-α5 extend out of the bifurcation point as a coiled-coil (CC1) followed by two HBs, HB7, and HB8. HB7 consists of H3-α4–5 and H5-α4, whereas HB8 contains H3-α1–3 and H5-α1–3 (Fig. 5c). HB9 comprising H3-α11–12 and H5-α11–13 initiates the long leg, which is followed by two CCs, CC2 (H3-α12 and H5-α13) and CC3 (H3-α13 and H5-α14). HB10 composed of three helices (H3-α14-15 and H5-α15) ends the long leg.

H1 and H4 form a stable heterodimer (Fig. 5f–h). The N-termini form a dumbbell-shaped structure, where HB11 (H1-α1–3 and H4-α1–2) and HB12 (H1-α2–3 and H4-α3–6) are connected by CC4 (H1-α2 and H4-α3). Their C-termini contain three CCs, CC5 (H1-α4 and H4-α7), CC6 (H1-α5 and H4-α8), and CC7 (H1-α6 and H4-α9). The H1/4 dimer holds H3/5's N and C-termini in both legs together at the bifurcation point, contacting HB9 in the long leg and CC1 in the short leg (Fig. 5a, c, f). Continuing downward, the H1/4 dimer joins the H3/5 C-termini in the long leg.

### Structural flexibility

Our cryo-EM analyses demonstrate augmin's highly flexible and dynamic nature. As shown in Fig. 5i, the neck moves relative to the legs as seen in distinct 2D classes, whereas each leg moves independently. The movements may occur around the loops between HBs, primarily in the H3/5 dimer, which we term 'joints' (Fig. 5c). Starting at the top of the neck, joint 1 (J1) formed of the α7–α8 loop of H3 (a.a. 226–251) and the α9–α10 loop H5 (a.a. 267–291) allows for movement at the HB4–5 connection (Fig. 5c–e). Similarly, J2 (H3 a.a. 181–199; H5 a.a. 166–182) creates movement within the middle of the neck, whereas J3 (H3 a.a. 428–445; H5 a.a. 453–478) allows alteration of the distance between the short and long legs. J4–5 and J6–7 allow for independent movements of the short and long legs, respectively.

## Discussion

In this study, we revealed the molecular architecture of human augmin using a combination of cryo-EM, computational methods, and CLMS. Augmin consists of a V-shaped head representing a functional tetramer of H2/6/7/8 and a tail created by H1/3/4/5. HAUS subunits in augmin primarily interact by forming CCs and HBs. We identified considerable flexibility in augmin, which most likely results from loops (or 'joints') between CCs and HBs. Our structural model can be extrapolated to other model organisms such as *X. laevis*, *Arabidopsis thaliana*, and *D. melanogaster* due to the structural and functional conservation of augmin. Though sequence identity and similarity are relatively low for some of the

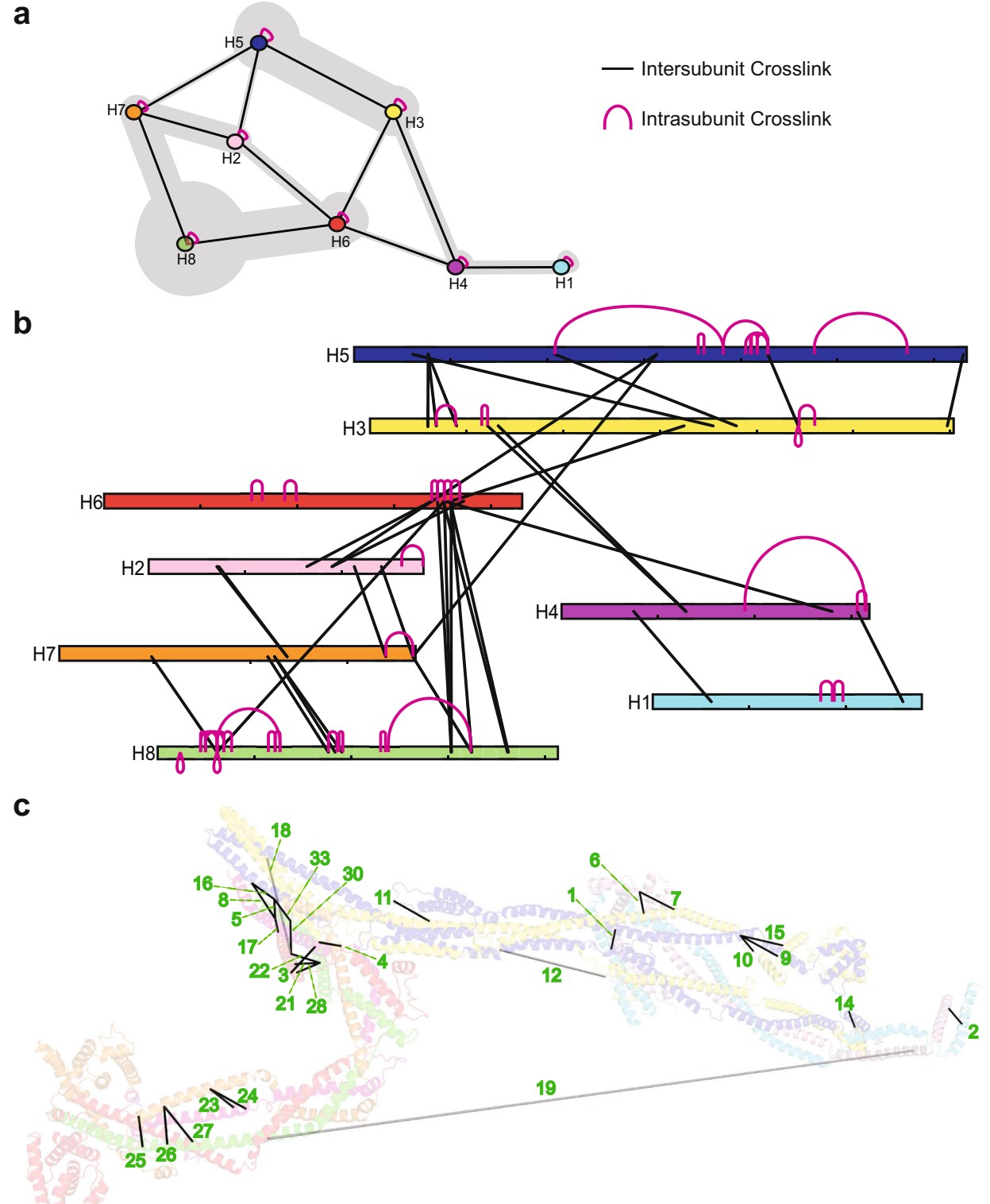

**Fig. 3 | Crosslinking mass spectrometry of augmin<sup>ΔH6C</sup>. a** Summary of crosslinks between HAUS subunits. The thickness of gray shading indicates the number of crosslinks. **b** Relative positions of each individual intrasubunit (pink) and intersubunit (black) crosslink identified by CLMS. **c** Intersubunit crosslinks are mapped onto the augmin structure using PyXlinkViewer. Subunits are colored as in panels (**a**), (**b**). Labels correspond to numbers in Table 1. Black lines represent crosslink Cα–Cα distances <30 Å and light gray lines represent distances >30 Å.

HAUS subunits across species (Supplementary Tables 3 and 4), AlphaFold2 predicted similar structures for both the H1/4 and H3/5 dimers as well as the head and tail across species (Fig. 6). Of note, the structure predictions allowed us to determine the human homologs of several augmin subunits of *D. melanogaster* that was,

to our knowledge, not previously reported. For instance, human H1, H2, H4, and H7 are homologs of *D. melanogaster* WAC, Msd1, Dgt2, and Msd5, respectively. Consistent with this, the predicted *D. melanogaster* WAC/Dgt2 combination shares a similar structure to that of human H1/4 (Fig. 6). Likewise, the V-shaped head of both

**Table 1 | Intersubunit crosslinks identified in purified augmin^ΔH6C complex**

| | CL distance (Å) | Protein 1 | Peptide 1 | Protein 2 | Peptide 2 | Score |
|---|---|---|---|---|---|---|
| 1 | 9.6 | HAUS1 | 60QKASEYESEAK | HAUS4 | 73LHKTTWLR | 13.2 |
| 2 | 13.7 | HAUS1 | 255IEEAKRELDSIEAELTR | HAUS4 | 301QATENKR | 20.4 |
| 3 | 16.8 | HAUS2 | 202NIPHLAANLKK | HAUS6 | 330LTVDLHYLEKETK | 43.3 |
| 4 | 10.6 | HAUS2 | 214KMNQALAK | HAUS7 | 335AVETVKK | 28.4 |
| 5 | 9.1 | HAUS2 | 241KQQNEVSSCIPK | HAUS7 | 362MNELMEK | 21.3 |
| 6 | 14.4 | HAUS3 | 122CQLMASVTSHKSLR | HAUS4 | 116ARLQQEVEEQLKKK | 21.1 |
| 7 | 17.9 | HAUS4 | 116ARLQQEVEEQLKKK | HAUS3 | 121NKCQLMASVTSHK | 32.4 |
| 8 | 19.4 | HAUS5 | 309STLLKER | HAUS2 | 241KQQNEVSSCIPK | 21.3 |
| 9 | 21.7 | HAUS5 | 77KLELEAAVTR | HAUS3 | 67SGKPILEGAALDEALK | 23.5 |
| 10 | 20.0 | HAUS5 | 77KLELEAAVTR | HAUS3 | 86TSDLKTPR | 18.5 |
| 11 | 21.8 | HAUS5 | 196AQFLQNLLLPQAKR | HAUS3 | 370QELVLNQLIKQK | 28.3 |
| 12 | 57.4 | HAUS5 | 428KVVPTFEAVAPQSR | HAUS3 | 339NTIDTKDYSTHR | 20.4 |
| 13 | a | HAUS5 | 465HRPGELKPLPTVLPSIHQLHPASPR | HAUS3 | 339NTIDTKDYSTHR | 37.3 |
| 14 | 10.5 | HAUS5 | 619LRWVQAQGALQKLCS | HAUS3 | 596IKAVSLED | 20.3 |
| 15 | 22.8 | HAUS5 | 77KLELEAAVTR | HAUS3 | 67SGKPILEGAALDEALK | 23.5 |
| 16 | 17.0 | HAUS5 | 309STLLKER | HAUS7 | 362MNELMEK | 26.3 |
| 17 | 21.1 | HAUS6 | 367HSVVEKQGEWHKK | HAUS2 | 241KQQNEVSSCIPK | 17.4 |
| 18 | 51.8 | HAUS6 | 358IKDDLTTIR | HAUS3 | 323LEKEVTQIK | 18.3 |
| 19 | 314.3 | HAUS6 | 217KMEPYDDHSNMEEKIQKVR | HAUS4 | 276QVLNSYEVLGEEFDR | 20.1 |
| 20 | b | HAUS6 | 348LSDLKHMR | HAUS8 | 60GKMSEGGR | 15.7 |
| 21 | 11.2 | HAUS6 | 348LSDLKHMR | HAUS8 | 299DVTAKK | 25.5 |
| 22 | 18.6 | HAUS6 | 358IKDDLTTIR | HAUS8 | 299DVTAKK | 22.5 |
| 23 | 11.9 | HAUS7 | 231QLQESAAKLHALR | HAUS2 | 113NLEIELLKLEK | 31.4 |
| 24 | 16.2 | HAUS7 | 231QLQESAAKLHALR | HAUS2 | 121LEKDTADVVHPFFLAQK | 16.4 |
| 25 | 14.5 | HAUS7 | 205QSDDWQWASASAKSEEEEKLAELAR | HAUS8 | 173RAEKNLLIMCK | 19.3 |
| 26 | 15.7 | HAUS7 | 219SEEEEKLAELAR | HAUS8 | 177NLLIMCKEK | 32.4 |
| 27 | 18.1 | HAUS7 | 219SEEEEKLAELAR | HAUS8 | 185EKLQKK | 18.4 |
| 28 | 11.7 | HAUS8 | 299DVTAKK | HAUS6 | 343FQKER | 20.3 |
| 29 | a | HAUS8 | 356ESGGAPKNTPLSEDDNPGASSAPAQATFISPSEDFSSSSQAEVPPSLSR | HAUS6 | 348LSDLKHMR | 31.4 |
| 30 | 16.5 | HAUS8 | 311SFAQVLELSAEASKEAALANQEVWEETQGMAPPSR | HAUS6 | 358IKDDLTTIR | 18.5 |
| 31 | a | HAUS8 | 356ESGGAPKNTPLSEDDNPGASSAPAQATFISPSEDFSSSSQAEVPPSLSR | HAUS6 | 355YRIKDDLTTIR | 25.3 |
| 32 | a | HAUS8 | 60GKMSEGGR | HAUS7 | 94FSSLKGVPTEVK | 17.5 |
| 33 | 20.1 | HAUS8 | 311SFAQVLELSAEASKEAALANQEVWEETQGMAPPSR | HAUS7 | 362MNELMEK | 26.5 |

Numbers in left column correspond to labels in Fig. 3c. Crosslinked amino acids are underlined in bold. Score is the Total XL score from Metamorpheus.
aPeptide 1 is not present in structure.
bPeptide 2 is not present in structure.

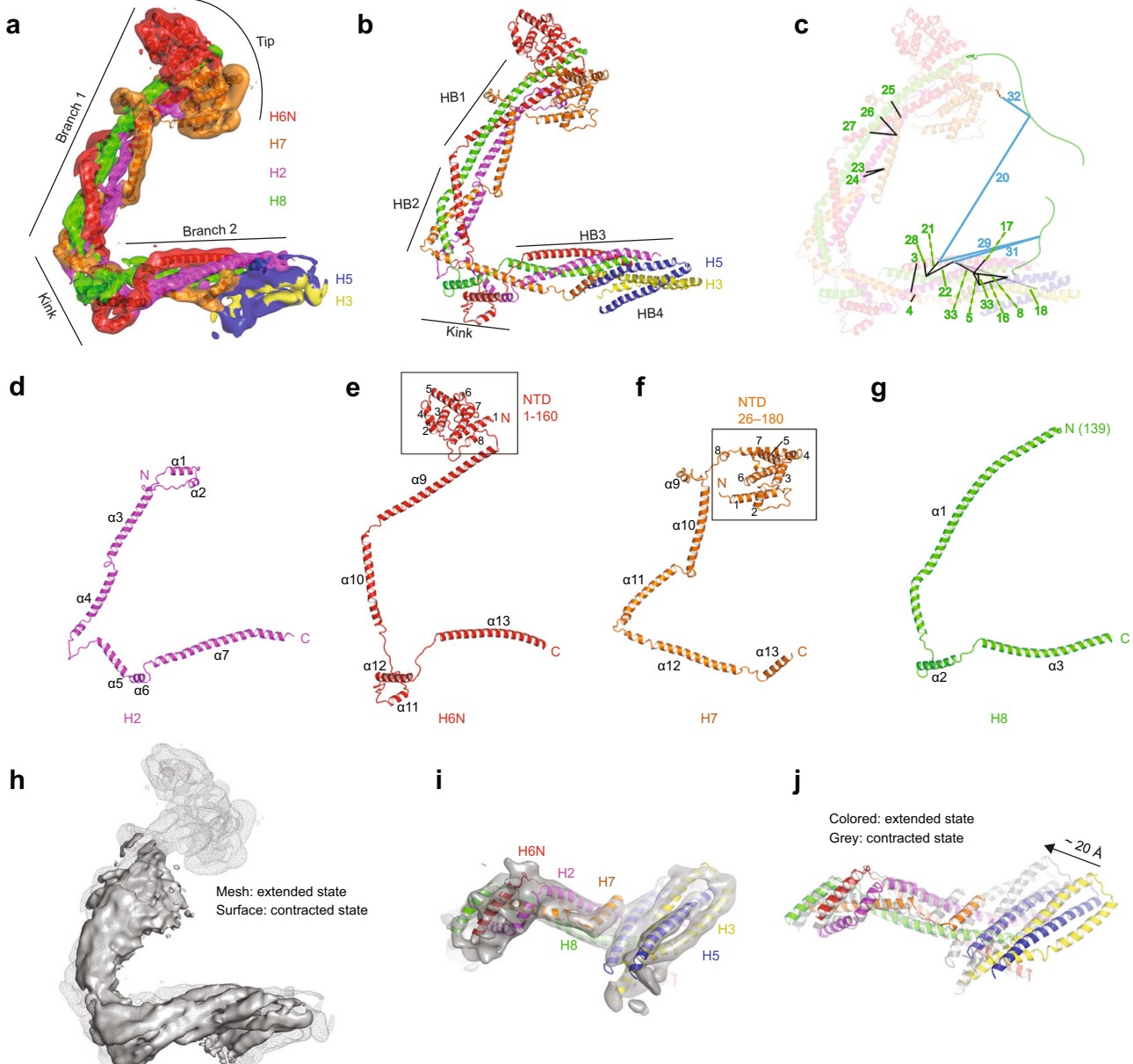

**Fig. 4 | Structure of the V-shaped head of augmin. a** Structure of the V-shaped head in extended conformation superimposed into the cryo-EM map with each subunit color-coded. **b** Structure of the V-shaped head in extended conformation in cartoon presentation. **c** Intersubunit crosslinks as labeled in Table 1 are mapped on the structure. Intersubunit crosslinks below 30 Å (black), above 30 Å (light gray), and those with one peptide from the N- and C-termini of H8 (drawn as green disordered chains) that are not modeled within augmin's structure (blue). **d** H2 with labeled helices numbered from N to C-terminus (α1–7). **e** H6N with labeled α-helices numbered from N to C-terminus (α1–13). NTD (a.a. 1–160, α1–8) enclosed within a box. **f** H7 with labeled α-helices numbered from N to C-terminus (α1–13). NTD (a.a. 1–160, α1–8) enclosed within a box. **g** H8 with labeled α-helices numbered from N to C-terminus (α1–3). **h** Overlayed cryo-EM maps of the extended state (mesh, gray) and contracted state (surface, gray). **i** Bottom view of the V-shaped head in the contracted state. **j** Structural comparison of the V-shaped head in the extended and contracted state. A conformational shift of -20 Å in the contracted state (gray) compared to the extended state (colored) is observed.

species are similar, although the *D. melanogaster* V-shaped head adopts a more extended state.

The N-terminus (a.a. 1–140) of H8, located at the tip of branch 1 within the V-shaped head, is vital for microtubule binding[27,28]. Therefore, the V-shaped head probably coordinates augmin's microtubule binding function. This region of H8 is not observed in the cryo-EM map, consistent with CD data showing that it forms a random coil in solution[29] and AlphaFold predictions suggesting an unstructured segment. Interestingly, post-translational modifications (PTMs) particularly phosphorylation have been identified in this region. These may regulate the affinity of augmin to microtubules[27,28].

Augmin and γ-TuRC association is driven by H6C[10,23], although the N-termini of H3/Dtg3 (a.a. 1–350) and H5/Dtg5 (a.a. 1–450) are also implicated in increasing augmin's affinity for γ-TuRC via NEDD1/Dgp71WD in *D. melanogaster*[8]. Sequence alignment of human and *D. melanogaster* subunits showed that human H3 (a.a. 1–331) and H5 (a.a. 1–378) are roughly equivalent to their counterparts in *D. melanogaster*. The N-termini of H3/5 encompass the entire short leg and most of the neck. Thus, the short leg and neck may be involved in augmin–γ-TuRC association via NEDD1. Taken together, the surface of augmin required for γ-TuRC recruitment through NEDD1 possibly comprises the head–tail connection down to the short leg. The V-shaped head of augmin would bind pre-existing microtubules while the γ-TuRC rests

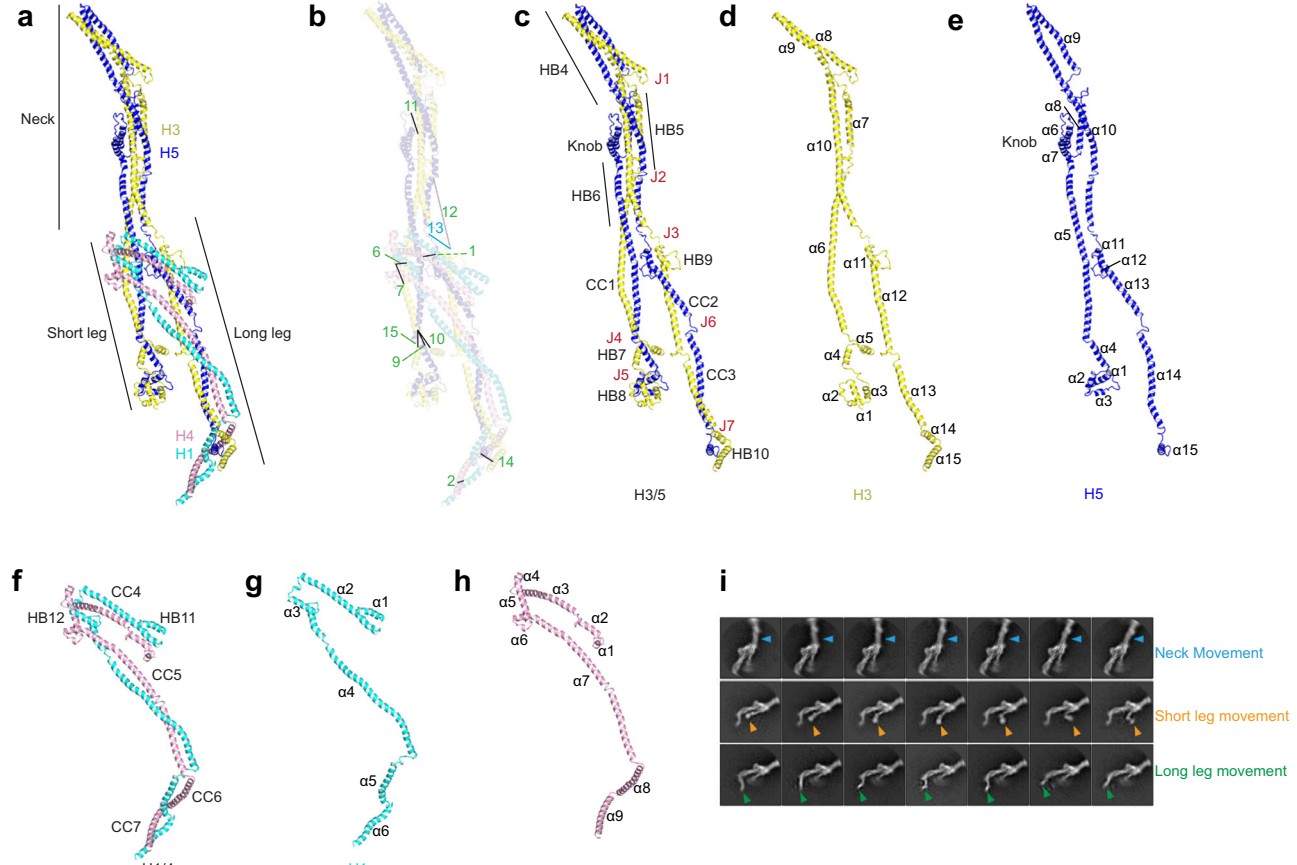

**Fig. 5 | Structure of the tail of augmin. a** Overall architecture of the tail of augmin with helices from H1/3/4/5 depicting the neck, short, and long legs. **b** Intersubunit crosslinks as labeled in Table 1 are mapped on the structure. Intersubunit crosslinks below 30 Å (black), above 30 Å (light gray), and one containing a peptide from a H5 region not modeled within augmin's structure (blue). **c** H3/5 dimer with labeled features: CC1–3, HB4–8, J1–7, and the knob. **d** H3 with labeled α-helices numbered

from N to C-terminus (α1–15). **e** H5 with labeled α-helices numbered from N to C-terminus (α1–15). **f** H1/4 dimer with labeled features: CC4–7 and HB11–12. **g** H1 with labeled α-helices numbered from N to C-terminus (α1–6). **h** H4 with labeled α-helices numbered from N to C-terminus (α1–9). **i** 2D classes depicting independent movement of the neck and the short and long legs.

against the side of augmin. As such, movement between the head and the tail, paired with augmin's flexibility would explain the observed angle variability of branching microtubules[4,11,30].

Our study utilized advanced computational approaches to help determine the position of different subunits within a cryo-EM map of a highly dynamic complex. However, these computational approaches alone are not yet sufficiently reliable to generate an accurate model for augmin. First, the prediction of single subunits within a protein complex (e.g. H1–8) lacked sufficient accuracy. Differences between the helix positioning from the AlphaFold Protein Structure Database compared to cryo-EM-based orientations can be seen in Supplementary Fig. 8. Prediction accuracy by ColabFold and AlphaFold Multimer tends to improve with the addition of interacting subunits or domains, especially when this is based on prior knowledge. We were not able to predict the full augmin structure due to computational resource limitations. However, based on our prediction of the V-shaped head and the tail, it might be challenging to predict an accurate structure for a large protein complex such as augmin. Thus, experimental data such as cryo-EM maps are likely required to build correct structures, and in our study, CLMS was needed to validate the subunit interfaces with confidence.

Our study shows that a combination of cryo-EM and advanced protein structure prediction can be harnessed to elucidate the structures of highly flexible, and dynamic protein complexes such as augmin. Furthermore, our structure, by delineating the subunit organization of augmin, provides insights into how augmin may

interact with both microtubules and the γ-TuRC, allowing for a more mechanistic approach to understanding branching microtubule nucleation. Future structural studies of the branching microtubule machinery will help to bridge gaps in our knowledge of the conserved processes mediating spindle assembly in mitosis and meiosis.

## Methods

### Molecular cloning and expression

The coding sequences for human HAUS subunits (H1–8) (isoforms can be found in Supplementary Table 3) were purchased from Integrated DNA Technologies (IDT), amplified by PCR, and cloned into pFBDM and pUCDM duet transfer plasmids using the USER cloning method[31]. Oligonucleotide sequences for PCR amplification before insertion into pFBDM and pUCDM can be found in Supplementary Table 5. Sequences for a double StrepII-tag together with a TEV cleavage site were attached to the N-terminus of H2. Three duet pUCDM plasmids were made while only two duet pFBDM were constructed. Specifically, duet pUCDM plasmids with H1 and H5 (pU_H15), H2 and H6N (pU_H26N), and H2 and H6 (pU_H26) were made while dueting pFBDM plasmids with H3 and H4 (pF_H34) and H7 and H8 (pF_H78) were constructed. Transfer plasmids pU_H15 and pF_H78 were transformed sequentially into DH10MultiBac[Cre] bacterial cells[31] to create a BACMID. BACMIDs were transfected into Sf9 (clonal isolate from IPLB-Sf-21-AE, Expression Systems, LLC, Cat. # 94-001F) insect cells to generate baculovirus with the genes for H1/5/7/8. A similar method was used to produce baculoviruses with genes for H2/3/4/6N or H2/3/4/6.

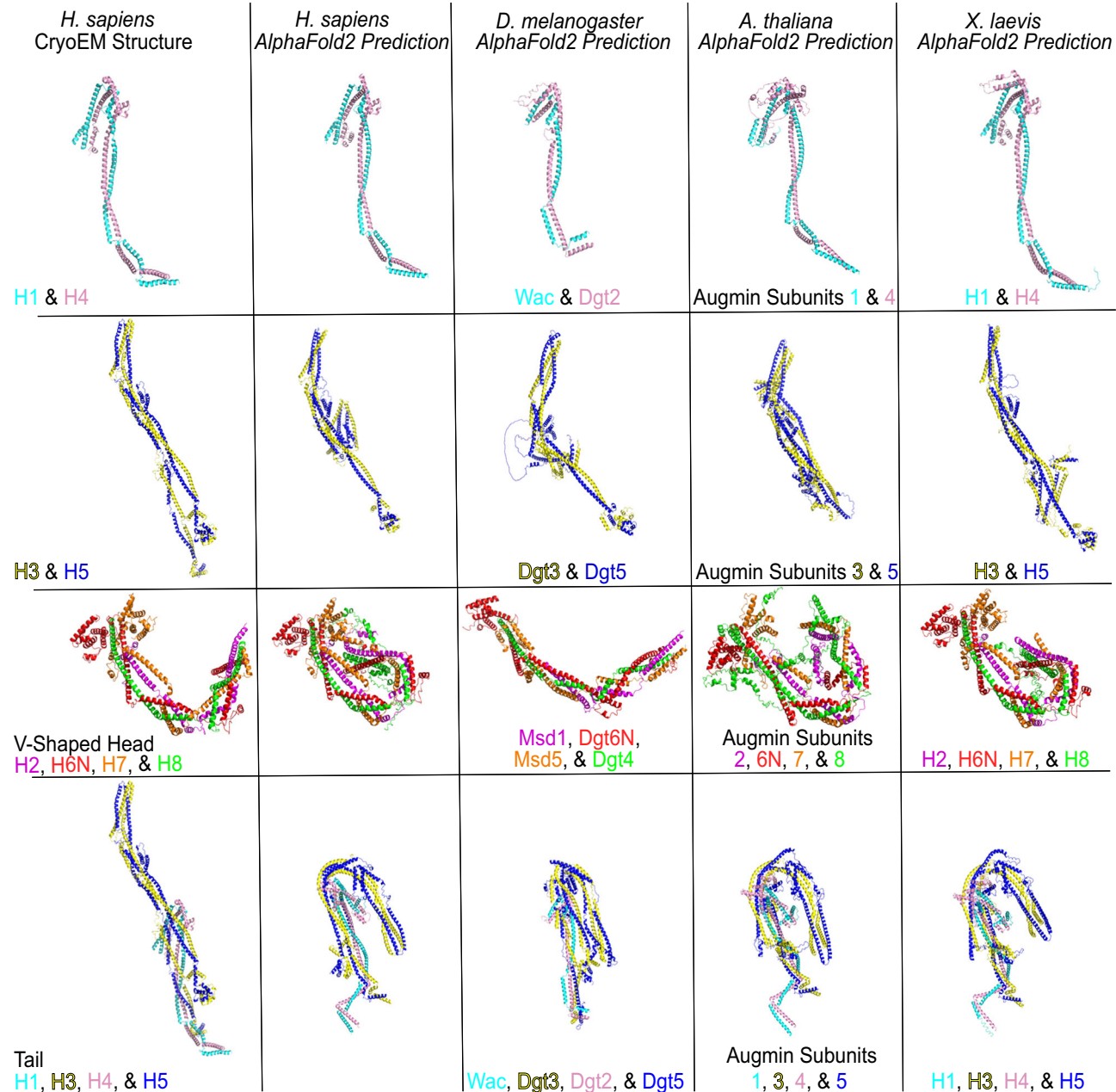

**Fig. 6 | Cross-species comparisons of the augmin complex predicted by AlphaFold2.** Comparisons of the H1/4 and H3/5 dimers, V-shaped head, and tail of augmin across *H. sapiens*, *D. melanogaster*, *A. thaliana*, and *X. laevis*. Cryo-EM structures of the *H. sapiens* H1/4 and H3/5 dimers and V-shaped head and tail regions are provided as references.

Infectious titers of all three baculoviruses were measured before infection. After viral titration, H1/5/7/8 and H2/3/4/6N (or H2/3/4/6) baculoviruses were used to co-infect High5™ (BTI-TN-5B1–4, Thermo Fisher Cat. # B85502) insect cells. After co-infection, augmin octamers were allowed to express for 2 days before cells were pelleted, flash frozen in liquid nitrogen, and stored for later use at −80 °C.

## Purification

Pelleted cells were resuspended in lysis buffer (40 mM HEPES, pH 8.0, 250 mM KCl, 2 mM DTT, 2 mM benzamidine, 1 mM EDTA, 5% glycerol, 0.1% Tween-20®, and Roche cOmplete™ EDTA-free Protease Inhibitor Cocktail). Cells were lysed and centrifuged for 1 h at 48,380 × *g* in a Beckman Coulter Avanti JXN-26 centrifuge. Cleared lysate was filtered using filter syringes tips before being loaded onto a StrepII-Trap® HP Column (Cytiva). Augmin complexes were eluted with lysis buffer

supplemented with 2.5 mM D-desthiobiotin. Augmin was concentrated and further purified over a Superose 6 column (Cytiva) in 20 mM HEPES, pH 8.0, 200 mM KCl, and 0.5 mM TCEP. Samples were further concentrated for cryo-EM and CLMS. Typically, 200 μl of augmin$^{\Delta H6C}$ at a concentration of 0.1–0.2 mg/ml was achieved from 3.0 l of insect cell culture. For the holo-complex, ~80 μl of sample at a similar concentration was obtained. Sample quality decreases dramatically after freezing and thawing; thus, freshly purified samples were always used for cryo-EM or CLMS.

## Electron microscopy

Freshly purified augmin samples were first visualized by negative-staining EM to check the sample quality. Grid screening revealed that on grids with no support or graphene oxide flakes, no monodisperse particles were observed (Supplementary Fig. 1d, e). However, those

with a thin, amorphous carbon layer exhibited Y-shaped particles, presumably intact augmin$^{\Delta H6C}$ (Supplementary Fig. 1f). Therefore, for data collection, aliquots of 3 µl samples at -0.05 mg/ml were applied onto glow-discharged Quantifoil R2/2 or R3.5/1 holey carbon grids covered with thin amorphous carbon film. The grids were incubated with the sample for 30 s at 4 °C and 100% humidity, blotted for 8 s and plunged into liquid ethane using an FEI Vitrobot.

Augmin$^{\Delta H6C}$ was imaged by a Titan Krios microscope (FEI) running at 300 kV equipped with a Gatan K2 summit detector, a GIF Quantum energy filter operated with a slit width of 20 eV, and a Volta Phase Plate. In total, 20,021 micrographs were collected using Leginon[41] in super-resolution mode at a nominal magnification of 105,000 (with a calibrated physical pixel size of 1.384 Å/pixel) with a fixed defocus value of −600 nm. Each micrograph was exposed for 8 s at a dose rate of 6 electron/pixel/s and saved as 40 movie frames. Calculated defocus values are in a range of −0.6 to −1.2 µm.

### Image processing

Motion correction was done with MotionCor2[42] within Appion[43]. Micrographs with poor quality were filtered out by visual inspection. Selected micrographs were imported into RELION[44] for further processing. CTF estimation was done with Gctf[45] while template-based auto-picking was done with RELION. Particles were extracted with a box size of ~500 Å that just covers the whole complex to avoid including neighboring particles. Multiple rounds of 2D classification were performed in cryoSPARC[46] and RELION. After cleaning, 1,173,445 particles remained.

Ab initio reconstruction and heterogeneous refinement with multiple classes in cryoSPARC were used to analyze the heterogeneity. By extensive classification, a complete map was determined at a ~20 Å resolution, from which the head and tail are visible (Fig. 1e).

An in-house python script named "rockstar.py", which allows for manual re-centering of particles based on 2D class averages was used to extract various regions (i.e. the head, legs, and neck) of each particle using a smaller box size. Specifically, in a RELION 2D classification job, each particle (location defined by coordinates rlnCoordinate$X$, rlnCoordinate$Y$) is aligned to a class average with in-plane translation offset (rlnOrigin$X$ and rlnOrigin$Y$) and in-plane rotation angle (rlnAngle$Psi$). By displaying a 2D average image by *relion_display*, a user can click the center of a region of interest (e.g. the head). The offsets of the region of interest relative to the center of the original image (d$x$, d$y$) are recorded. Based on these parameters and the equations below, new coordinates ($X_{new}$, $Y_{new}$) with the region of interest as the center are calculated for each particle. A new particle extraction is done with the new coordinates and smaller box size to only extract the region of interest. The source code of "rockstar.py" is available on GitHub (https://github.com/zhuangli200/Rockstar).

$$X_{new} = \text{rlnCoordiante}X - \text{rlnOrigin}X + dx * \cos(\text{rlnAngle}Psi/180*\pi) + dy* \sin(\text{rlnAngle}Psi/180*\pi) \tag{1}$$

$$Y_{new} = \text{rlnCoordinate}Y - \text{rlnOrigin}Y + dy* \cos(\text{rlnAngle}Psi/180*\pi) - dx* \sin(\text{rlnAngle}Psi/180*\pi) \tag{2}$$

We segmented augmin into three representative regions, including the V-shaped head, neck in the tail, and legs in the tail. After ab initio reconstruction and heterogeneous refinement in cryoSPARC, the V-shaped head was classified into three classes with two conformational states, the extended (classes 1 and 3) and contracted states (class 2) (Supplementary Fig. 2d). Further refinement was performed in RELION to obtain the segmented maps shown in Fig. 1e and Supplementary Fig. 2d, f, h.

All resolution estimations were based on the gold-standard Fourier shell correlation (FSC) calculations using the FSC = 0.143 criteria in RELION. A summary of EM data collection and refinement is listed in Supplementary Table 1.

To generate a complete map of augmin, individual maps of the V-shaped head in an extended conformation, the legs, and the neck were fitted into a lower resolution map of augmin (Fig. 1e) and merged into one map using the *vop maximum* program in Chimera[47].

### Structure prediction, subunit assignment, and sequence alignment

AlphaFold predictions of single HAUS subunits were downloaded from the AlphaFold Protein Structure Database[32,35]. Structures of binary combinations of HAUS subunits (full lengths for all subunits except for H6) were predicted using the ColabFold Notebook[33]. H6 was shortened to the H6N construct to fit within the 1400 a.a. limit of the notebook and to match the protein construct used in cryo-EM analysis.

For the prediction of more than two subunits using ColabFold, specific regions of HAUS subunits are fused by a 60-glycine linker. For example, to predict the structure of Branch 1, H2(a.a. 1–278)–60 glycine–H8(a.a. 101–410) and H6(a.a. 1–269)–60 glycine–H7(1–299) are provided as the two chains. For the kink, H2(a.a. 115–155)–60 glycine–H8(a.a. 247–284) and H6(a.a. 264–432)–60 glycine–H7-(300–368) were used for structure prediction as two chains. To complete the V-shaped head, branch 2 was predicted as a single chain with H2(a.a. 158–235)–60 glycine–H6(a.a. 331–432)–60 glycine– H7(a.a. 300–368)–60 glycine–H8(a.a. 284–410). For the legs, two different models were used. For the short leg and neck, H3(a.a. 1–432) and H5(a.a. 1–457) were used as separate chains. The long leg was predicted using H1(a.a. 1–278)–60 glycine–H4(a.a. 1–318) and H3(a.a. 433–603)–60 glycine–H5(a.a. 458–633). Structure predictions were then used for modeling in COOT[36] to be described below.

Upon the release of AlphaFold Multimer[34], binary combinations were again completed using a local GPU workstation for comparison with the ColabFold predictions for completeness. No major differences were observed. Additional combinations of the V-shaped head (H2/6N/7/8) and the tail (H1/3/4/5) were also completed in AlphaFold Multimer for *H. sapiens*, *D. melanogaster*, *A. thaliana*, and *X. laevis*.

Sequence identity and similarity across *H. sapiens*, *D. melanogaster*, *A. thaliana*, and *X. laevis* were calculated using a BLOSUM62 matrix through Snapgene (SnapGene software from Insightful Science) and then compared to outputs from Clustal $\Omega$[48]. Sequence alignments between species were completed using the MPI Bioinformatics Toolkit[49].

### Model building

Structure predictions from ColabFold combinations described above were fit within the cryo-EM map using rigid body fitting in Chimera. Specifically, ColabFold predictions of the long leg (Supplementary Fig. 5c) and the neck and short leg (Supplementary Fig. 5d) were fit within the tail while those of branch 1 (Supplementary Fig. 5e), the kink (Supplementary Fig. 5f) and branch 2 (Supplementary Fig. 5g) were fit to the V-shaped head. Overlapping structures and loops were removed and sequences were connected within COOT. After sequence connections were completed in COOT to piece all of the subunits back together from the different models from ColabFold, refinement of the structure model against the cryo-EM map was performed using the *phenix:real_space_refine* tool in Phenix[50].

### CLMS analysis

Purified augmin complex samples were pooled and subjected to crosslinking using DSSO at different concentrations in molar excess to augmin. Reactions of DSSO:augmin at 250:1, 500:1, 750:1, and 1000:1 molar ratios were incubated on ice for 1 h before quenching with 1.0 M Tris, pH 8.0. A control reaction of augmin with DMSO was subjected to

the same experimental conditions. The sample was split with some samples being kept in solution while the rest was run on an SDS–PAGE gel.

In-gel digestion was performed as described[51] with slight modifications. Briefly, ~2.5 μg crosslinked protein was electrophoresed by SDS–PAGE and stained with Coomassie Blue. High molecular weight protein bands were excised and destained with 50% acetonitrile in 25 mM ammonium bicarbonate. Proteins were reduced with 10 mM DTT for 30 min at 25 °C followed by alkylation with 55 mM chloroacetamide for 1 h in the dark at 25 °C. Samples were dehydrated with 100% acetonitrile, rehydrated in 20 μg/ml TrypZean® (MilliporeSigma) in 50 mM ammonium bicarbonate, and incubated for ~16 h at 37 °C. Peptides were extracted twice by adding acetonitrile to 60% and incubating 10 min, then desalted on Pierce C18 spin tips (Thermo Scientific, PI84850) and dried by vacuum centrifugation prior to MS analysis.

For gel-free analysis, crosslinked samples were supplemented with 6 M urea (from 8 M stock), 5 mM tris(2-carboxyethyl) phosphine, and 30 mM chloroacetamide to a final volume of 200 μl and incubated 1 h at 37 °C. Samples were diluted with 3 volumes (600 μl) of fresh 50 mM ammonium bicarbonate to reduce urea concentration (<2 M), supplemented with 0.5 μg TrypZean®, and incubated at 37 °C. After 12 h, a second aliquot of 0.5 μg TrypZean® was added and incubation continued for an additional 12 h. Digestion was quenched with 0.1% trifluoroacetic acid to a final volume of 850 μl. Peptides were desalted using Pierce C18 spin columns (Thermo Scientific, PI89870) and dried under vacuum.

Tryptic peptides were solubilized in 3% acetonitrile/0.1% formic acid and analyzed by reverse phase LC–ESI–MS/MS using a Dionex UltiMate 3000 RSLCnano system coupled to an Orbitrap Fusion Lumos mass spectrometer (Thermo Fisher Scientific, Waltham, MA). Peptides were loaded on a trap column (300 μm ID × 5 mm) packed with 5 μm particle/100 Å pore PepMap C18 resin at 5 μl/min. They were resolved on an Aurora UHPLC emitter column (25 cm × 75 μm ID) packed with 1.6 μm/120 Å C18 resin (Ionopticks, Victoria, Australia) at a constant flow rate of 200 nl/min and a temperature of 40 °C. LC solvent A was 0.1% formic acid and solvent B was 0.1% formic acid/80% acetonitrile. The analytical column was developed with an 80 min linear gradient from 8% to 27% solvent B, 20 min from 27% to 45% B, 5 min from 45-100% B. The Lumos was operated in positive ion and data-dependent acquisition modes using the advanced peak determination function. MS1 and MS2 scan ranges were 375–1500 and 300–1250 $m/z$, respectively. Precursor ions were fragmented by higher energy collision dissociation at a normalized collision energy setting of 30%. Orbitrap resolution was 120,000 and 7500 for MS1 and MS2, respectively, with maximum injection time of 50 ms for MS1 and 20 ms for MS2. Dynamic exclusion was set at 60 s with 10 ppm tolerance. Identification of DSSO-crosslinked peptides was performed with MetaMorpheus V0.0.320[38]. Raw data from a DMSO-treated control sample and a DSSO crosslinked sample were searched together against a custom database containing the sequences of the 8 subunits of the recombinant augmin complex using the three-stage MetaMorpheus workflow: calibration, general peptide search, and crosslink search. We applied both strict (Lys–Lys crosslinks only) and relaxed (Lys/Ser/Thr/Tyr as potential crosslink sites) crosslinker specificity based on prior results from CLMS of the *D. melanogaster* augmin complex[8]. We only show data from the relaxed search since it included all crosslinks identified in the Lys–Lys search. MetaMorpheus search parameters are provided in Supplementary Dataset 1. To create the final list of unique crosslinked peptide pairs, we first concatenated the search results from all crosslinking and sample prep conditions (250:1, 500:1, 750:1, 1000:1 DSSO:augmin molar ratio, and in-gel vs. in-solution digests). Our

MetaMorpheus search settings resulted in 1% FDR for both intersubunit and intrasubunit crosslink pairs when calculated using an established formula for target-decoy searches, FDR = (TD−DD)/TT, where TT is the number of target–target crosslink pairs, TD is the number of target–decoy crosslink pairs, and DD is the number of decoy–decoy crosslink pairs[52,53]. We then eliminated redundant entries, keeping the highest scoring incidences, and applied a final cutoff below MetaMorpheus $q$-value of 0.01 (1% FDR). Crosslinked peptides were visualized using the xVis software[54] and PyXlinkViewer[55]. The mass spectrometry data have been deposited to the ProteomeXchange Consortium (http://proteomecentral.proteomexchange.org) via the PRIDE partner repository[56] with the dataset identifier PXD031411. A concatenated, unfiltered results file containing the intrasubunit and intersubunit crosslinks identified by MetaMorpheus is provided in Supplementary Dataset 1.

### Map visualization
Figures were generated using PyMOL, Chimera[47], and ChimeraX[57].

### Reporting summary
Further information on research design is available in the Nature Research Reporting Summary linked to this article.

## Data availability
Cryo-EM reconstruction of the augmin complex has been deposited in the Electron Microscopy Data Bank under accession number EMD-25387. Coordinates for the atomic model of augmin have been deposited in the Protein Data Bank under accession number 7SQK. Mass spectrometry data has been deposited in the PRIDE Database under submission number PXD031411. Single protein predictions for HAUS1–8 are available from the AlphaFold Protein Structure Database (https://alphafold.ebi.ac.uk/). Uncropped gels are provided as a Source Data file with this paper. Source data are provided with this paper.

## Code availability
The source code of "rockstar.py" is available on GitHub (https://github.com/zhuangli200/Rockstar).

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

## Acknowledgements

We thank Thomas Klose for help with cryo-EM and Steven Wilson for computation. We thank Uma Aryal and Jackeline Franco from the Purdue Proteomics Facility for assistance with LC–MS data collection, and Wen Jiang and Daisuke Kihara for helpful discussions. This work made use of the Purdue Cryo-EM Facility. We thank both facilities for their help. L.C. is supported by the Department of Biological Sciences at Purdue University, the National Institutes of Health (NIH) [R01GM138675], and a Showalter Trust Research Award; C.G. is supported by a grant from the NIH [T32GM132024], and the Purdue Ross-Lynn Fellowship. Mass spectrometry analysis was made possible with support from the Indiana Clinical and Translational Sciences Institute which is funded in part by Award Number UL1TR002529 from the National Institutes of Health, National Center for Advancing Translational Sciences, Clinical and Translational Sciences Award (M.C.H). D.B. is supported by UKRI/Medical Research Council MC_UP_1201/6 and Cancer Research UK C576/A14109.

## Author contributions

L.C., M.C.H., and D.B. supervised the study. Z.Z. cloned the augmin complex. C.G., J.Y., and L.C. prepared samples. C.G., Z.L., and L.C. collected and processed cryo-EM data. C.G. and L.C. performed all computational analyses using AlphaFold2. C.G., A.G.D., and M.C.H. performed the CLMS sample preparation and data analysis. All authors analyzed the data. All authors contributed to the manuscript preparation.

## Competing interests

The authors declare no competing interests.
