## [Peer Review File · Nature Communications]

REVIEWER COMMENTS

Reviewer #1 (Remarks to the Author):

In the manuscript "Molecular architecture of the augmin complex", authors reported the first atomic structure model of human augmin complex, an eight-subunit protein complex which play a critical role in microtubule nucleation. This model would contribute to the understanding of the molecular mechanism of branching microtubule nucleation.

Due to the flexibility of the structure of augmin, the achieved resolution of the cryo-EM data was limited, however taking advantage of structure prediction tools, authors could generate models for interacting subunits of augmin and eventually fit these subunits into the observed EM density and arrived at a model with atomic resolution. Furthermore, authors validated the structure model using orthogonal experimental data generated using crosslinking mass spectrometry.

The combination of the technique is sensible and demonstrated a new possible path towards high resolution structure models of proteins/protein complexes. However, crosslinking mass spectrometry analysis should be improved before the manuscript should be published in Nature Communications.

In this study, the number of crosslinks identified from the augmin complex is fairly low. Reported inter-protein crosslinks only cover a small portion of interface between subunits. For validating the structure model, the dataset is rather thin. As the authors are working with recombinant protein complex, sample amount if not really a restriction. Authors could improve the density/coverage of crosslinking data by improving the depth of analysis or using alternative crosslinker such as heterobifunctional crosslinker SDA or sulfo-SDA.

The claim on the 0% FDR for identified crosslinks is conceptionally wrong. Assuming that the authors took the score cut-off where the first decoy match started to appear, then according to the number of crosslinks reported, the link level FDR% could be estimated between 3-6% for inter-protein crosslinks and 2.4-4.8% for intra-protein crosslinks.

There are eight crosslinks (over 10%) disagree with the structure model. The distance between linked residues in these crosslinks are far beyond the crosslinking limit. However, there was no explanation provided on possible cause of these crosslinks. The SDS-PAGE image shown in Fig S6 indicated two major crosslinking products (two clear gel bands), each containing similar proportion of material. How did authors determine that the selected upper gel band was monomeric crosslinked augmin? Authors stated in the manuscript that there is an alternative conformation for the V-shaped head. Is the second gel band (with a lower MS) corresponding to a different conformation? In the in-solution digested sample, these crosslinking products were analysed as a mixture. Are over-length crosslink only present in the in-solution digested sample or they are also present in the data from the isolated gel band?

Similarly, there are three self-links identified from HAUS2 and HAUS7, indicating crosslinking also captured oligomers of the complex and crosslinks form between molecules are abundant enough to be detected. When all crosslinking products were analysed as a mixture (i.e the in-solution digested sample), one can not confidently tell if an observed crosslink is from a single molecule of augmin complex or formed between two molecules of the complex (expect for self-links with overlapped peptide sequences). In such case, one would benefit from isolating the monomeric crosslinked augmin, for example using SDS-PAGE, before digestion and LC-MS analysis.

For the LC-MS/MS analysis, authors applied a rather low maximum injection time for MS2 scans (20 ms), for typically low abundant crosslinked peptides, this could lead to poor quality of MS2 spectra. Although, the authors have deposited the crosslinking mass spectrometry data in PRIDE Database, they did not provide reviewer access to the dataset. Therefore, no judgement can be made on the quality of actual MS data.

It was stated that "MetaMorpheus search parameters are provided in Supplemental Dataset 1", however, information could not be found in the provided file. Some key parameters used for database search, such as MS1 and MS2 error tolerance, missed cleavages, fixed and variable modifications etc., should be included in the methods section or supplemental data.

Reviewer #2 (Remarks to the Author):

Using a combination of cryo-electron microscopy, alphaFold protein structure predictions and crosslinking mass spectroscopy, the authors propose the first atomic model of human augmin complex - despite its flexibility -, being able to assign all 8 augmin subunits. Together with the available literature information about interactions between the various augmin subunits and augmin's interaction partners, the microtubule and gammaTuRC, this structure sheds new light onto the mechanism of branched microtubule nucleation.

Given the specific expertise of this reviewer it seems appropriate to comment mostly on the protein expression and purification and here the main comment is that not sufficient detail is provided to reproduce expression and purification.

In previous publications of recombinant augmin, only very small amounts of purified protein could be obtained, given its apparently fairly insoluble nature. The authors should therefore state their yields so that they can be compared to previous attempts and provide more detail about how their expression construct was constructed, the expression and purification conditions, particularly if a major advance was made here in purifying the complex as it seems.

The authors should also clarify which isoforms for the subunits were used. It seems that for the holocomplex the longest HAUS6 isoform was used with an expected molecular weight of 108.6kDa (table "Subunit names..." and scheme in Fig. S1), but this subunit runs with a molecular weight that is lower than the 94kDa marker in the Coomassie gel in Fig. 1B. Please explain.

From the scheme, it seems that the longest HAUS4 isoform was used, but in the table it is stated that the isoform 4 was used which is by 45 amino acids shorter. Please clarify.

It remained also unclear to this reviewer which resolution the various reconstructions have.

Reviewer #3 (Remarks to the Author):

Gabel et al present an impressive study, where strengths of multiple tools (cryo-EM, in silico structure prediction and crosslinking mass spec validation) are synergised to reconstruct the augmin complex. Execution of every step on the complex path to this novel and important multimeric structure is sound and I enthusiastically support publication.

Specific points:

It is intriguing how such a tight-knit complex (inter-connecting helical bundles) would be unstable, which precluded use of normal holey grids. Could binding energy for the whole complex be reliably predicted/computed by any program? Or would any program indicate how the complex might be expected to dissociate? i.e. into what sub-complexes? Would the previously identified tetramers that the authors refer to be expected among the predicted dissociation products?

The authors eventually choose thin continuous carbon support for their sample, but also show graphene in the Supplement without any discussion. Perhaps a bit of explanation in the Methods or at least in the S1 figure legend would be useful, since it is not that easy to interpret the example micrographs shown.

Why did authors bother to truncate H6? Was it only because it is longer than other subunits? Not necessarily a problem even if entirely disordered (the complex appeared similar in micrographs with or without the H6C tail). There were other disordered tails, too. Was it deleted because otherwise it would not fit to the sequence limit for the structure prediction program? Perhaps it should be stated a bit more clearly.

In the text describing crosslinks shown in Fig. 3C (end of page 5), it would be helpful to quote crosslink numbers in brackets for each discussed case/type of crosslink to aid cross-referencing.

Perhaps crosslinking that the authors optimised would enable higher resolution structures of various states (dropping continuous carbon and the phase plate). Not suggesting that for this paper, but could the authors comment? Do you think it could be a viable approach for the future?

Minor points:

Fig 1A does not include NEDD1, although the Fig callout directly follows its mention in the main text, so I expected to see some cartoon representation of it in that panel. Is not enough known about NEDD1's position in the complex to include it or would it be expected to be hidden when viewed from this angle or is this simply for clarity? Perhaps the callout should be moved somewhere else or the figure or legend updated to clarify that.

The Head and Neck labels are a bit confusing in Fig 1E. Please, consider placing next to the part they describe.

Szymon W. Manka

REVIEWER COMMENTS

Reviewer #1 (Remarks to the Author):

In the manuscript “Molecular architecture of the augmin complex”, authors reported the first atomic structure model of human augmin complex, an eight-subunit protein complex which play a critical role in microtubule nucleation. This model would contribute to the understanding of the molecular mechanism of branching microtubule nucleation.

Due to the flexibility of the structure of augmin, the achieved resolution of the cryo-EM data was limited, however taking advantage of structure prediction tools, authors could generate models for interacting subunits of augmin and eventually fit these subunits into the observed EM density and arrived at a model with atomic resolution. Furthermore, authors validated the structure model using orthogonal experimental data generated using crosslinking mass spectrometry.

The combination of the technique is sensible and demonstrated a new possible path towards high resolution structure models of proteins/protein complexes. However, crosslinking mass spectrometry analysis should be improved before the manuscript should be published in Nature Communications.

1. In this study, the number of crosslinks identified from the augmin complex is fairly low. Reported inter-protein crosslinks only cover a small portion of interface between subunits. For validating the structure model, the dataset is rather thin. As the authors are working with recombinant protein complex, sample amount if not really a restriction. Authors could improve the density/coverage of crosslinking data by improving the depth of analysis or using alternative crosslinker such as heterobifunctional crosslinker SDA or sulfo-SDA.

Re: We agree with the reviewer that there are parts of the complex that lack crosslinking coverage. There are a number of possible reasons for this such as lack of accessible pairs of crosslinking groups in these regions. We wish to point out that the crosslinking efforts are intended to test if the overall arrangement of subunits in the complex is correct. For that purpose, we were primarily concerned with the fraction of high confidence crosslinks that are consistent with the structural model, instead of generating extensive coverage of the subunit interaction interfaces.

During the revision period, we have put extensive efforts into improving the coverage of the crosslinking data in several ways as described below.

1. We relaxed the search criteria in MetaMorpheus in various ways to determine if we could find convincing crosslinks mapping to augmin regions lacking crosslink coverage. Although we could find additional spectra matching the existing identified crosslink pairs, this effort did not reveal new reliable crosslink matches. Therefore, we think that our search criteria in MetaMorpheus are appropriate for our model-testing purpose.
2. We evaluated our crosslinking datasets using another well-established search program created by the Rappaport lab, xiSEARCH, that uses a different method for finding and scoring crosslinked peptide pairs. This search gave a partially overlapping set of intersubunit crosslinked peptide IDs. However, there were no unique crosslink pairs from xiSEARCH in regions of the augmin model not already represented by the MetaMorpheus hits. 100% of the interlinked peptide pairs identified by both MetaMorpheus and xiSEARCH analyses had crosslink distances $<30 \text{ \AA}$ and were therefore consistent with the structural model (see the following table and Figures R1 and

R2 below). Because the additional xiSEARCH analysis did not improve the coverage of the augmin model significantly, we decided not to include it in the revised manuscript for the sake of simplicity.

- We have tried several times to perform additional crosslinking experiments with the heterobifunctional crosslinker sulfo-LC-SDA (ThermoFisher Cat. #26174; armlength: ~ 12.5 Å), which has a similar crosslinking distance as DSSO. While we can efficiently crosslink augmin based on SDS-PAGE analysis, we have been unable to identify high-scoring crosslinked peptide pairs using either MetaMorpheus or xiSEARCH. We can convincingly identify many individual peptides bearing the crosslinker modification, but for reasons that are unclear, we are unable to identify high-scoring intra- or inter-subunit crosslinked peptide pairs with either program.
- We also tried the crosslinker sulfo-SDA (ThermoFisher Cat. #26173; armlength: ~ 3.9 Å) which contains a shorter spacer arm than sulfo-LC-SDA. The search results came out similar to that of sulfo-LC-SDA.

Table of Identified Inter-subunit Crosslinks from xiSEARCH with crosslink distances <30 Å

	CL Distance (Å)	Protein 1	Peptide 1	Protein 2	Peptide 2	xiSEARCH score
1	9.6	Haus1	⁶⁰ Q KASEYESEAK	Haus4	⁷³ LH K TTWLR	12.1
2	13.7	Haus1	²⁵⁵ IEEAK R ELDSIEAELTR	Haus4	³⁰¹ QATEN K R	15.0
3	10.6	Haus2	²¹⁴ K MNQALAK	Haus7	³³⁵ AVETV K K	21.4
4	9.1	Haus2	²⁴¹ K QQNEVSSCIPK	Haus7	³⁶² MNELME K	13.8
5	16.8	Haus2	²⁰² NIPHLAANL K K	Haus6	³³⁰ LTVDLHYLE K ETK	15.7
6	17.9	Haus4	¹¹⁶ ARLQQEVEEQ LK K	Haus3	¹²¹ N K CQLMASVTSHK	21.9
7	10.5	Haus5	⁶¹⁹ LRWVQAQGALQ K LCS	Haus3	⁵⁹⁶ I K AVSLED	11.7
8	17.0	Haus5	³⁰⁹ STLL K ER	Haus7	³⁶² MNELME K	16.9
9	24.9	Haus5	⁵⁵ T V K KIR	Haus3	⁸³ TCKTSD L KTPR	15.7
10	21.8	Haus5	¹⁹⁶ AQFLQNLLLPQA K R	Haus3	³⁷⁰ QELVLNQLI K QK	16.8
11	13.2	Haus5	³⁷⁸ ELQA K QQR	Haus3	³⁴² ENAQLLNMPVVKGDFDL QIA K QDYTTAR	19.1
12	20.0	Haus5	⁷⁷ K LELEAAVTR	Haus3	⁸⁶ TSD L KTPR	14.0
13	22.8	Haus5	⁷⁷ K LELEAAVTR	Haus3	⁶⁷ S G KPILEGAALDEALK	12.3
14	19.4	Haus5	³⁰⁹ STLL K ER	Haus2	²⁴¹ K QQNEVSSCIPK	16.9
15	11.9	Haus7	²³¹ QLQESAA K LHALR	Haus2	¹⁰⁵ NLEIELLKLEK	17.2
16	11.2	Haus6	³⁴⁸ LSD L K HMR	Haus8	²⁹⁹ DVT A K K	21.2
17	18.6	Haus6	³⁵⁸ I K DDLTIR	Haus8	²⁹⁹ DVT A K K	15.8
18	18.1	Haus7	²¹⁹ SEEE E K LAEAR	Haus8	¹⁸⁵ E K LQ K K	15.3
19	20.1	Haus8	³¹¹ SFAQVLELSAEAS K EALAN QEVWEETQGMAPPSR	Haus7	³⁶² MNELME K	15.5
20	15.7	Haus7	²¹⁹ SEEE E K LAEAR	Haus8	¹⁷⁷ NLLIMC K EK	16.0

- Crosslinked amino acids are underlined in bold
- Crosslinks identified at 5% FDR

Figure R1: xiSEARCH Xlink Map for comparison (only Xlinks <30 angstroms are shown).

Figure R2: MetaMorpheus Xlink Map from Fig. 3c.

2. The claim on the 0% FDR for identified crosslinks is conceptionally wrong. Assuming that the authors took the score cut-off where the first decoy match started to appear, then according to the number of crosslinks reported, the link level FDR% could be estimated between 3-6% for inter-protein crosslinks and 2.4-4.8% for intra-protein crosslinks.

Re: We recognize that our choice of words may be technically incorrect. However, our final list of reported peptide pairs was filtered to remove redundancies so the actual FDR is probably much lower than what it appears by the reviewer's calculation method. To avoid any confusion or misrepresentation of the FDR we have edited the wording in our manuscript to clearly indicate that we used the position of the first decoy match as our score cutoff to ensure we only included high confidence matches (thus 0% of the hits are from the decoy database). We no longer claim a 0% FDR in the text (revision of the relevant sentence is shown below; lines 161–162).

To maximize the rigor of this analysis, we used the position of the first decoy match as our score cutoff to ensure we only included high confidence matches we only considered peptide pairs identified with scores above the 0% false discovery rate (FDR) threshold.

3. There are eight crosslinks (over 10%) disagree with the structure model. The distance between linked residues in these crosslinks are far beyond the crosslinking limit. However, there was no explanation provided on possible cause of these crosslinks.

Re: The augmin complex is highly flexible and conformationally heterogeneous based on our cryo-EM analysis. Thus, we think that those crosslinks may reflect distinct conformations of augmin, or the inherent flexibility and natural motion of the complex allows distant regions to come into proximity. We included the discussion in the section 'Crosslinking Mass Spectrometry' (also pasted below; lines 176–

177). Nonetheless, the majority of identified crosslinks are supportive of the overall architectural arrangement of subunits in augmin.

'Only 3 of the 33 intersubunit crosslinks are inconsistent with the model based on predicted Ca-Ca distances (crosslinks 18 (54 Å), 12 (59 Å), and 19 (312 Å) in Fig. 3c and Table 1), though complex flexibility and different conformational states of augmin may possibly explain these data points.'

The SDS-PAGE image shown in Fig S6 indicated two major crosslinking products (two clear gel bands), each containing similar proportion of material. How did authors determine that the selected upper gel band was monomeric crosslinked augmin? Authors stated in the manuscript that there is an alternative conformation for the V-shaped head. Is the second gel band (with a lower MS) corresponding to a different conformation? In the in-solution digested sample, these crosslinking products were analysed as a mixture. Are over-length crosslink only present in the in-solution digested sample or they are also present in the data from the isolated gel band?

Re: We apologize for the arrow on the gel in Fig. S6, which was misleading and was not intended to indicate we were selecting only the larger band. We don't have any good way of knowing which crosslinked gel band corresponds to a single intact augmin complex, so we combined all of the high molecular weight material in our in-gel analysis (the lower crosslinked band could be only a partial complex, for example). We recognize that this may result in some background of crosslinks between augmin complexes, but we decided it was more practical to combine all high MW species. Almost all of the final data that we report is from the in-gel digested samples, so the few long-distance crosslinks are also present in the gel-separated samples.

4. Similarly, there are three self-links identified from HAUS2 and HAUS7, indicating crosslinking also captured oligomers of the complex and crosslinks form between molecules are abundant enough to be detected. When all crosslinking products were analysed as a mixture (i.e the in-solution digested sample), one can not confidently tell if an observed crosslink is from a single molecule of augmin complex or formed between two molecules of the complex (expect for self-links with overlapped peptide sequences). In such case, one would benefit from isolating the monomeric crosslinked augmin, for example using SDS-PAGE, before digestion and LC-MS analysis.

Re: We agree with everything the reviewer has stated here. However, our attempts to isolate monomeric crosslinked augmin were unsuccessful. On SDS-PAGE, we tried various cross-linking conditions (adjusting concentrations of crosslinker and reaction time); however, we were not able to identify a pure population of crosslinked monomeric complex.

5. For the LC-MS/MS analysis, authors applied a rather low maximum injection time for MS2 scans (20 ms), for typically low abundant crosslinked peptides, this could lead to poor quality of MS2 spectra. Although, the authors have deposited the crosslinking mass spectrometry data in PRIDE Database, they did not provide reviewer access to the dataset. Therefore, no judgement can be made on the quality of actual MS data.

Re: Our intention was to have the PRIDE submission available to the reviewers, and we have made sure it is now publicly accessible. The data is now available in the PRIDE database under PXD031411. We also note that we have two representative annotated crosslink spectra in Figure S6 that demonstrate the typical quality of the MS2 spectra. It is difficult for us to change MS method parameters because we are using a core facility for instrument access that services our entire campus research community. Nonetheless, we have manually inspected many of the matched crosslinked peptide spectra in Table 1 to ensure the assignments are realistic.

6. It was stated that “MetaMorpheus search parameters are provided in Supplemental Dataset 1”, however, information could not be found in the provided file. Some key parameters used for database search, such as MS1 and MS2 error tolerance, missed cleavages, fixed and variable modifications etc., should be included in the methods section or supplemental data.

Re: We inadvertently submitted an earlier version of the Supplementary Dataset 1 that lacked the search parameters page and a couple other pages of information. We apologize for this error on our part. We have now included the correct Supplementary Dataset 1 file.

Reviewer #2 (Remarks to the Author):

Using a combination of cryo-electron microscopy, alphaFold protein structure predictions and crosslinking mass spectroscopy, the authors propose the first atomic model of human augmin complex - despite its flexibility -, being able to assign all 8 augmin subunits. Together with the available literature information about interactions between the various augmin subunits and augmin's interaction partners, the microtubule and gammaTuRC, this structure sheds new light onto the mechanism of branched microtubule nucleation.

Given the specific expertise of this reviewer it seems appropriate to comment mostly on the protein expression and purification and here the main comment is that not sufficient detail is provided to reproduce expression and purification.

Re: This is very insightful. We left out some of the details of expression and purification to reduce the length of the methods section. We have added more detail for clarification ('Molecular Cloning and Expression' and 'Purification' in the Methods section; lines 395–424).

In previous publications of recombinant augmin, only very small amounts of purified protein could be obtained, given its apparently fairly insoluble nature. The authors should therefore state their yields so that they can be compared to previous attempts and provide more detail about how their their expression construct was constructed, the expression and purification conditions, particularly if a major advance was made here in purifying the complex as it seems.

Re: We have included more detail on expression and purification for clarity. We have also included more information about the yield of the augmin complex ('Purification' in the Methods section; lines 420–424).

The authors should also clarify which isoforms for the subunits were used. It seems that for the holocomplex the longest HAUS6 isoform was used with an expected molecular weight of 108.6kDa (table "Subunit names..." and scheme in Fig. S1), but this subunit runs with a molecular weight that is lower than the 94kDa marker in the Coomassie gel in Fig. 1B. Please explain.

Re: We have clarified which isoforms by stating it before their NCBI accession numbers in Supplementary Table 3. We are unsure why H6 runs lower than its predicted molecular weight. Proteins can sometimes run differently than their predicted molecular weight on SDS-PAGE gels. This is a 4–20% gradient gel to help clarify, but we are not sure why it runs at this position.

From the scheme, it seems that the longest HAUS4 isoform was used, but in the table it is stated that the isoform 4 was used which is by 45 amino acids shorter. Please clarify.

Re: We thank the reviewer for noticing this error. We have corrected the schematic in Supplementary Fig. 1 to the reflect isoform of H4 used in our studies.

It remained also unclear to this reviewer which resolution the various reconstructions have.

Re: We thank the reviewer for their comment. This information can be found in Supplementary Table 1 and the FSC Curves for the different reconstructions can be found in Supplementary Fig. 2i.

Reviewer #3 (Remarks to the Author):

Gabel et al present an impressive study, where strengths of multiple tools (cryo-EM, in silico structure prediction and crosslinking mass spec validation) are synergised to reconstruct the augmin complex. Execution of every step on the complex path to this novel and important multimeric structure is sound and I enthusiastically support publication.

Specific points:

It is intriguing how such a tight-knit complex (inter-connecting helical bundles) would be unstable, which precluded use of normal holey grids. Could binding energy for the whole complex be reliably predicted/computed by any program? Or would any program indicate how the complex might be expected to dissociate? i.e. into what sub-complexes? Would the previously identified tetramers that the authors refer to be expected among the predicted dissociation products?

Re: We appreciate the point raised by the reviewer. We agree that such a complex maintained by coiled-coil interactions becoming dissociated on holey carbon grids is unlikely. Y-shaped augmin particles not being observed on holey grids is more likely caused by other reasons. One possibility is that augmin complex tends to not enter the holes without a support layer¹. We have modified our description of our results to no longer use disassembled in the text (lines 85–87; also pasted below).

First, we used a thin, continuous carbon film to observe monodisperse Y-shaped particles, which otherwise were not seen on grids with no support or graphene oxide flakes (Supplementary Fig. 1d–f).

The authors eventually choose thin continuous carbon support for their sample, but also show graphene in the Supplement without any discussion. Perhaps a bit of explanation in the Methods or at least in the S1 figure legend would be useful, since it is not that easy to interpret the example micrographs shown.

Re: We thank the reviewer for their comment. We have added discussion in the methods section about screening of cryo-EM grids. Y-shaped monodisperse particles, presumably intact augmin^{ΔH6C}, were only observed on grids covered with thin amorphous carbon film but not on grids with no support or graphene oxide flakes ('Electron microscopy' in the Methods section; lines 428–431).

Why did authors bother to truncate H6? Was it only because it is longer than other subunits? Not necessarily a problem even if entirely disordered (the complex appeared similar in micrographs with or without the H6C tail). There were other disordered tails, too. Was it deleted because otherwise it would not fit to the sequence limit for the structure prediction program? Perhaps it should be stated a bit more clearly.

Re: We agree with the reviewer that the disordered region of H6C may not be a problem for cryo-EM, although H6C is quite long (> 500 amino acids). H6C is not a problem for structure prediction either. The major reason for truncation of H6C is because it improved augmin complex expression and purification yield, consistent with a previous study by Hsia *et al.*². This improvement in purification yield made it more practical for cryo-EM analysis as well as CLMS. We have added this sentence to clarify this point (first paragraph in the Results section; lines 76–78).

In the text describing crosslinks shown in Fig. 3C (end of page 5), it would be helpful to quote crosslink numbers in brackets for each discussed case/type of crosslink to aid cross-referencing.

Re: Thank you for your points. We have made these changes (beginning of page 6; lines 175–178).

Perhaps crosslinking that the authors optimised would enable higher resolution structures of various states (dropping continuous carbon and the phase plate). Not suggesting that for this paper, but could the authors comment? Do you think it could be a viable approach for the future?

Re: The inherent flexibility of augmin is a major limitation factor in higher resolution determination for this complex. Crosslinking may not be able to overcome this limitation. We think a better solution would be to use a binding partner of augmin (NEDD1, the full γ -TuRC, EML3, etc.) to make the complex less flexible to improve resolution. We are currently working in this direction.

Minor points:

Fig 1A does not include NEDD1, although the Fig callout directly follows its mention in the main text, so I expected to see some cartoon representation of it in that panel. Is not enough known about NEDD1's position in the complex to include it or would it be expected to be hidden when viewed from this angle or is this simply for clarity? Perhaps the callout should be moved somewhere else or the figure or legend updated to clarify that.

Re: We originally left out NEDD1 because its exact binding site on γ -TuRC and to augmin are postulated but not specifically known, so we chose not to include it in Fig. 1A to avoid any controversy in its placement. After careful consideration, we have added a cartoon representation of NEDD1 for completeness of the model. Thank you for your insight.

The Head and Neck labels are a bit confusing in Fig 1E. Please, consider placing next to the part they describe.

Re: We have moved the labels with respect to the reviewer's comment in Fig. 1e.

Szymon W. Manka

- 1 Snijder, J. *et al.* Vitrification after multiple rounds of sample application and blotting improves particle density on cryo-electron microscopy grids. *J Struct Biol* **198**, 38-42, doi:10.1016/j.jsb.2017.02.008 (2017).
- 2 Hsia, K. C. *et al.* Reconstitution of the augmin complex provides insights into its architecture and function. *Nat Cell Biol* **16**, 852-863, doi:10.1038/ncb3030 (2014).

REVIEWERS' COMMENTS

Reviewer #1 (Remarks to the Author):

In the revised manuscript, authors have addressed a few questions/comments from the reviews and improved the manuscript. However there are still few issues regarding to how authors presented the crosslinking/MS data. These issues need to be addressed before the data is published.

1) Original comments: The claim on the 0% FDR for identified crosslinks is conceptually wrong. Assuming that the authors took the score cut-off where the first decoy match started to appear, then according to the number of crosslinks reported, the link level FDR% could be estimated between 3-6% for inter-protein crosslinks and 2.4-4.8% for intra-protein crosslinks.

Author response: We recognize that our choice of words may be technically incorrect. However, our final list of reported peptide pairs was filtered to remove redundancies so the actual FDR is probably much lower than what it appears by the reviewer's calculation method.

>> In fact, removing redundancy will likely make the problem rather worse than better. Depending on what redundancy has been removed you are more likely to remove more true positives than false positives and therefore, at least in relative terms, make the error larger.

Author response: To avoid any confusion or misrepresentation of the FDR we have edited the wording in our manuscript to clearly indicate that we used the position of the first decoy match as our score cutoff to ensure we only included high confidence matches (thus 0% of the hits are from the decoy database). We no longer claim a 0% FDR in the text. (revision of the relevant sentence is shown below; lines 161–162).

To maximize the rigor of this analysis, we used the position of the first decoy match as our score cutoff to ensure we only included high confidence matches we only considered peptide pairs identified with scores above the 0% false discovery rate (FDR) threshold.

>>Typically, one estimates the probability of random matches (false matches) in a dataset based on the number of decoy matches passed the selected score cutoff. Making a score cutoff at the first appear decoy match provides no information on the actual level of confidence. Although one can somehow still calculate a potential error rate based on the number of matches with score higher than the first decoy, however this is a lot less accurate than a proper decoy-based FDR estimation. As the first occurrence of a decoy is somewhat stochastic – and the same is true for unknown false positives – you could have either passed no false positive matches, but just as likely have one or more false positives among your target matches.

Based on the number of crosslinks passed your first decoy cutoff, your data would likely have an error rate of >5% (after removing redundancy). It is fine to state “with such a cutoff, you did take matches with an as high as possible confidence. However, it is an overstate that “such a cutoff ensure you only included high confidence matches.”

It is more accurate to accept e.g., 5%FDR and then see if the discrepancy between your model and the crosslink data roughly agrees with the expected FDR. For example, in the manuscript, you have presented 3 out of 33 that are inconsistent with the model. Seeing your basically undefined FDR, these could, at least in part, be false positives. In some ways this makes the crosslinks rather agree more with your model than less.

2) There are eight crosslinks (over 10%) disagree with the structure model. Authors focused on three interprotein crosslinks and stated that these overlength crosslinks are because of flexibility of the complex. Authors should provide further data on structural flexibility (such as EM data) to support this conclusion. For example, crosslink 19 linked the two ends of the complex, was there any EM evidence showed possible conformation of the complex in which Tip and the end of long leg touch. When evaluating the agreement between the crosslinking data and the structural model, intra-protein crosslinks should not be left out, especially there were also a bigger proportion (5 out of 42) of intra-protein crosslinks violated the crosslinking limit.

Authors also should not exclude the possibility that these over-length crosslinks have resulted from false matches, or crosslinks formed between two copies of the complex molecules. As pointed in the original review comments, there are crosslinks indicating the existence of homo-multimeric crosslinks of the complex.

3) During the revision work, authors have tried to increase the density/coverage of their crosslinking data. However, with additional experiment efforts using SDA based crosslinkers, authors did not get additional crosslinks. Very likely this was because the diazirine end of crosslinker give rise to a lot less linking specificity, which lead to more diverse and further diluted crosslinking products in comparison to DSSO, without additional efforts on enrichment of the crosslinked peptides, the depth of the MS analysis is very limited.

The author's pointed out that “the crosslinking efforts are intended to test if the overall arrangement of subunits in the complex is correct.” As the current data showed acceptable confidence level, it is fair to state that at the regions where crosslinks have been observed, they show reasonable agreement with the structural model. However, due to the relative low coverage of the crosslinking data, the sentence in the manuscript “Collectively, the CLMS data strongly support the overall accuracy of our atomic model of augmin.” should be down tuned.

Additional point: As crosslinking/MS is still a relatively new technique in comparison to EM or X-ray crystallography, it would be good to include a reference to the technique, for example a review.

Reviewer #2 (Remarks to the Author):

The authors have adequately addressed my concerns, with one exception: could they please also state the yield of the non-truncated holocomplex ("less than half" is fairly meaningless). Together with the other improvements this is a very nice manuscript providing interesting information.

REVIEWERS' COMMENTS

Reviewer #1 (Remarks to the Author):

In the revised manuscript, authors have addressed a few questions/comments from the reviews and improved the manuscript. However there are still few issues regarding to how authors presented the crosslinking/MS data. These issues need to be addressed before the data is published.

1) Original comments: The claim on the 0% FDR for identified crosslinks is conceptually wrong. Assuming that the authors took the score cut-off where the first decoy match started to appear, then according to the number of crosslinks reported, the link level FDR% could be estimated between 3-6% for inter-protein crosslinks and 2.4-4.8% for intra-protein crosslinks.

Author response: We recognize that our choice of words may be technically incorrect. However, our final list of reported peptide pairs was filtered to remove redundancies so the actual FDR is probably much lower than what it appears by the reviewer's calculation method.

>> In fact, removing redundancy will likely make the problem rather worse than better. Depending on what redundancy has been removed you are more likely to remove more true positives than false positives and therefore, at least in relative terms, make the error larger.

Author response: To avoid any confusion or misrepresentation of the FDR we have edited the wording in our manuscript to clearly indicate that we used the position of the first decoy match as our score cutoff to ensure we only included high confidence matches (thus 0% of the hits are from the decoy database). We no longer claim a 0% FDR in the text. (revision of the relevant sentence is shown below; lines 161–162).

To maximize the rigor of this analysis, we used the position of the first decoy match as our score cutoff to ensure we only included high confidence matches we only considered peptide pairs identified with scores above the 0% false discovery rate (FDR) threshold.

>>Typically, one estimates the probability of random matches (false matches) in a dataset based on the number of decoy matches passed the selected score cutoff. Making a score cutoff at the first appear decoy match provides no information on the actual level of confidence. Although one can somehow still calculate a potential error rate based on the number of matches with score higher than the first decoy, however this is a lot less accurate than a proper decoy-based FDR estimation. As the first occurrence of a decoy is somewhat stochastic – and the same is true for unknown false positives – you could have either passed no false positive matches, but just as likely have one or more false positives among your target matches.

Based on the number of crosslinks passed your first decoy cutoff, your data would likely have an error rate of >5% (after removing redundancy). It is fine to state “with such a cutoff, you did take matches with an as high as possible confidence. However, it is an overstate that “such a cutoff ensure you only included high confidence matches.”

It is more accurate to accept e.g., 5%FDR and then see if the discrepancy between your model and the crosslink data roughly agrees with the expected FDR. For example, in the manuscript, you have presented 3 out of 33 that are inconsistent with the model. Seeing your basically undefined FDR, these could, at least in part, be false positives. In some ways this makes the crosslinkes rather agree more with your model then less.

Re: In retrospect we see the reviewer's point that selecting the position of the first decoy match as a cutoff is not appropriate because of the stochastic nature by which decoy matches will appear from one

experiment to another. We also agree with all of the reviewer's points about the nature of FDR measurements and the value of having them.

To address these concerns, we used a published method for empirically calculating FDR from CLMS experiments (applied to our compiled interlink and intralink search results, prior to removing redundancies). This information and the associated references have been added to the CLMS methods section now (page 18). In short, the calculated FDRs for our interlinked and intralinked peptide MetaMorpheus results were both $\sim 1\%$. After removing redundancies, we applied a final cutoff based on the MetaMorpheus-calculated q-value of 1% (0.01). This final cutoff removed only a few of the lowest-scoring peptide pairs. This achieved the same result as cutting off at the first decoy hit and did not require altering any of the data in our tables or figures.

We also edited the results text slightly on the bottom of page 5 to reflect the calculation of $\sim 1\%$ FDR from our MetaMorpheus search and removal of the statement about cutting off at the first decoy hit.

2) There are eight crosslinks (over 10%) disagree with the structure model. Authors focused on three interprotein crosslinks and stated that these overlength crosslinks are because of flexibility of the complex. Authors should provide further data on structural flexibility (such as EM data) to support this conclusion. For example, crosslink 19 linked the two ends of the complex, was there any EM evidence showed possible conformation of the complex in which Tip and the end of long leg touch. When evaluating the agreement between the crosslinking data and the structural model, intra-protein crosslinks should not be left out, especially there were also a bigger proportion (5 out of 42) of intra-protein crosslinks violated the crosslinking limit.

Authors also should not exclude the possibility that these over-length crosslinks have resulted from false matches, or crosslinks formed between two copies of the complex molecules. As pointed in the original review comments, there are crosslinks indicating the existence of homo-multimeric crosslinks of the complex.

Re: We agree with this assessment. On page 6 we have expanded how we address the small number of inconsistent crosslinks with the statement: "The few inconsistencies could reflect complex flexibility and different conformational states of augmin, crosslinks between multiple augmin complexes, or false positive search matches".

3) During the revision work, authors have tried to increase the density/coverage of their crosslinking data. However, with additional experiment efforts using SDA based crosslinkers, authors did not get additional crosslinks. Very likely this was because the diazirine end of crosslinker give rise to a lot less linking specificity, which lead to more diverse and further diluted crosslinking products in comparison to DSSO, without additional efforts on enrichment of the crosslinked peptides, the depth of the MS analysis is very limited.

The author's pointed out that "the crosslinking efforts are intended to test if the overall arrangement of subunits in the complex is correct." As the current data showed acceptable confidence level, it is fair to state that at the regions where crosslinks have been observed, they show reasonable agreement with the structural model. However, due to the relative low coverage of the crosslinking data, the sentence in the manuscript "Collectively, the CLMS data strongly support the overall accuracy of our atomic model of augmin." should be down tuned.

Re: To acknowledge the lack of complete coverage of the complex by CLMS data and tone down the strength of our conclusion we have modified the sentence in question to read "Collectively, the CLMS data support the overall accuracy of our atomic model of augmin, although some regions of the complex lacked CLMS data coverage".

Additional point: As crosslinking/MS is still a relatively new technique in comparison to EM or X-ray crystallography, it would be good to include a reference to the technique, for example a review.

Re: At the beginning of the CLMS results section (page 5) we added the sentence "CLMS is becoming a useful tool for supplementing protein structural characterizations" and provided a reference to a review article from this year on use of CLMS in supporting protein structural characterizations.

Reviewer #2 (Remarks to the Author):

The authors have adequately addressed my concerns, with one exception: could they please also state the yield of the non-truncated holocomplex ("less than half" is fairly meaningless). Together with the other improvements this is a very nice manuscript providing interesting information.

Re: We have removed "less than half"; instead, we stated the yield of the augmin holo-complex.